# Specificity of AMPylation of the human chaperone BiP is mediated by TPR motifs of FICD

Joel Fauser[1,2], Burak Gulen [1,2], Vivian Pogenberg [1], Christian Pett [3], Danial Pourjafar-Dehkordi[4], Christoph Krisp[5], Dorothea Höpfner [1,2], Gesa König[1], Hartmut Schlüter [5], Matthias J. Feige[2,6], Martin Zacharias[4], Christian Hedberg [3✉] & Aymelt Itzen [1,2,7✉]

To adapt to fluctuating protein folding loads in the endoplasmic reticulum (ER), the Hsp70 chaperone BiP is reversibly modified with adenosine monophosphate (AMP) by the ER-resident Fic-enzyme FICD/HYPE. The structural basis for BiP binding and AMPylation by FICD has remained elusive due to the transient nature of the enzyme-substrate-complex. Here, we use thiol-reactive derivatives of the cosubstrate adenosine triphosphate (ATP) to covalently stabilize the transient FICD:BiP complex and determine its crystal structure. The complex reveals that the TPR-motifs of FICD bind specifically to the conserved hydrophobic linker of BiP and thus mediate specificity for the domain-docked conformation of BiP. Furthermore, we show that both AMPylation and deAMPylation of BiP are not directly regulated by the presence of unfolded proteins. Together, combining chemical biology, crystallography and biochemistry, our study provides structural insights into a key regulatory mechanism that safeguards ER homeostasis.

[1] Department of Biochemistry and Signal Transduction, University Medical Center Hamburg-Eppendorf (UKE), Hamburg, Germany. [2] Center for Integrated Protein Science Munich (CIPSM), Department Chemistry, Technical University of Munich, Garching, Germany. [3] Chemical Biology Center (KBC), Institute of Chemistry, Umeå University, Umeå, Sweden. [4] Physics Department T38, Technical University of Munich, Garching, Germany. [5] Institute of Clinical Chemistry and Laboratory Medicine, Mass Spectrometric Proteomics, University Medical Center Hamburg-Eppendorf (UKE), Hamburg, Germany. [6] Institute for Advanced Study, Technical University of Munich, Garching, Germany. [7] Center for Structural Systems Biology (CSSB), University Medical Center Hamburg-Eppendorf (UKE), Hamburg, Germany. ✉email: christian.hedberg@umu.se; a.itzen@uke.de

Protein homeostasis (proteostasis) plays a critical role in the survival of cells. Deficiencies in proteostasis result in cell death and cause several diseases[1]. Since the endoplasmic reticulum (ER) is responsible for the correct folding of approximately one-third of all synthesized proteins in eukaryotic cells, proteostasis in the ER is highly regulated. The interconnection between the amount of newly synthesized proteins and the folding capacity of the ER is constantly maintained by numerous mechanisms. In the case of a high load of unfolded proteins, a coordinated signal cascade, the unfolded protein response (UPR), is launched to re-establish the equilibrium between folding load and ER folding capacity[2]. Global protein translation is downregulated to decrease the burden of unfolded proteins entering the ER, while the expression of chaperones and enzymes assisting protein folding are upregulated. The heat shock protein (Hsp) 70 family member BiP is a major chaperone within the ER that assists protein folding and degradation as well as contributes to UPR regulation[3,4]. Like all Hsp70s, BiP consists of two domains: an N-terminal nucleotide-binding domain (NBD) and a substrate-binding domain (SBD) followed by a C-terminal lid[5]. The NBD and the SBD are connected by a conserved hydrophobic linker. BiP interacts transiently with unfolded proteins via its SBD by binding to exposed hydrophobic amino acid (aa) stretches[6,7]. The SBD thereby shields the unfolded target from unspecific aggregation and links the substrate to the comprehensive machinery of BiP cochaperones that may promote folding or degradation[8]. BiP's action on its clients is governed by the ATPase activity of the NBD. The nucleotide adenosine triphosphate (ATP) can bind to the NBD and is hydrolyzed to adenosine diphosphate (ADP) by the intrinsic ATPase activity with the concomitant release of phosphate. During this ADP/ATP cycle, BiP adopts different conformations: in the ATP-bound state, BiP has low affinity to its substrates and its NBD and SBD are docked to each other with the conserved linker inserted into a specific pocket of the NBD[5]. In the ADP-bound state, BiP exhibits high affinity toward unfolded proteins while the NBD and SBD are undocked[9–11]. In vivo, transfer of folding clients and adoption of the high-affinity state is achieved by stimulation of ATP hydrolysis via J-domain-containing proteins (J-proteins), also referred to as Hsp40 cochaperones[12]. Furthermore, nucleotide exchange factors accelerate ADP/ATP exchange, thereby releasing substrates and restoring BiP in its client accepting ATP bound state[13].

In order to match ER folding capacities to short-term fluctuations of the unfolded protein load, two mechanisms directly regulate BiP: first, the oligomerization of BiP, and second, the covalent modification of BiP with an adenosine monophosphate (AMP) moiety[14–16]. This process is a posttranslational modification, referred to as AMPylation, in which an AMP is transferred from ATP to the protein side chains[17,18]. Recently, the AMPylated residue of BiP was linked to T518 within the SBD[19]. In consequence, the conformational equilibrium of AMPylated BiP is shifted toward the ATP-bound state and stimulation of ATP hydrolysis by J-proteins is impaired. Thus, AMPylation inhibits the chaperone activity of BiP[19,20]. The AMPylation of BiP is catalyzed by an ER-resident AMP transferase FICD (also known as HYPE), the only human representative of the family of filamentation induced by cyclic-AMP (Fic) enzymes. Fic enzymes consist of a structural core (Fic-fold) containing a conserved catalytic motif HxFxDGNGRxxR and are known to transfer nucleoside monophosphates to their dedicated targets[21,22]. FICD contains two N-terminal TPR motifs and a transmembrane helix that anchors the enzyme to the ER membrane[23]. While FICD was demonstrated to preferentially AMPylate ATP-bound BiP, the molecular basis for this preference remained unclear[19]. Of note, we and others suggested further potential AMPylation substrates

of FICD, such as the eukaryotic elongation factor 1A2 (EEF1A2) and uridine 5′ monophosphate synthase (UMPS)[24–26]. Furthermore, FICD can form a homodimer via its Fic domains[23]. Like many other Fic enzymes, FICD has an inhibitory α-helix containing the conserved sequence motif (S/T)xxxE(G/N) close to the ATP-binding site that keeps the enzyme intrinsically inactive in terms of AMP transfer[27,28]. Substitution of the conserved glutamate in the inhibitory motif by glycine (in FICD: E234G) was shown to stimulate AMPylation activity in vitro[27]. FICD wild type (WT), however, exhibits an antagonistic enzymatic activity that specifically removes the AMP from BiP (a process called deAMPylation). This dual enzymatic mode of some Fic enzymes has so far been reported for FICD and EfFic (from *Enterococcus faecalis*)[29,30]. Recently, it was demonstrated that AMPylation activity is also conferred by monomeric FICD (induced by a single-point mutation L258D, thus disrupting dimerization) despite the integrity of the inhibitory motif, suggesting the monomeric species as AMPylator in vivo[31,32].

While the individual structures of FICD and BiP are well characterized, the structural analysis of their interaction proved elusive[5,23]. In general, Fic enzymes exhibit low affinity for their targets[33–35]. Indeed, only one structure of a bacterial Fic-AMPylase in complex with its human substrate is available to date[36]. Very recently, we have reported a method that allows specific covalent linkage of Fic enzyme:substrate complexes by using thiol-reactive nucleotide derivatives (TReNDs)[26]. The method is based on the strategic substitution of an aa within the ATP-binding pocket of Fic enzymes by cysteine, which reacts with the TReND. The TReND-equipped Fic enzyme captures its substrate in a subsequent AMPylation reaction, thus forming a covalently linked ternary complex. While this concept was initially designed to identify novel targets of Fic enzymes, it has proven beneficial for structural analyses by stabilizing the interaction of Fic enzymes and other AMP transferases with their targets[26,37,38].

In order to gain insights into the fundamentals of FICD-mediated BiP AMPylation that directly regulates ER homeostasis, we here solve the atomic structure of the FICD:BiP complex by X-ray crystallography at 2.6 Å. We stabilize the transient interaction by covalent tethering of FICD to BiP by using cosubstrate analogs (TReNDs). Our findings demonstrate that FICD's TPR motifs are essential for FICD-mediated BiP AMPylation and confer specificity of FICD toward the ATP-bound state of BiP. Furthermore, the TPR motifs are also required for AMPylation of the previously suggested FICD substrate EEF1A2 but not for AMPylation of UMPS. Lastly, biochemical data suggest that (de)AMPylation of BiP is not directly regulated by binding to folding substrates.

## Results

**Formation of a covalent complex consisting of FICD and BiP.** To investigate the structural basis of the recognition of BiP by FICD, we used a method for covalent stabilization of this transient complex[26]. To this end, we engineered the ATP-binding pocket of FICD by a cysteine substitution. The engineered cysteine reacts with TReNDs, thus forming a binary adduct, referred to as FICD$^{TReND}$ (Fig. 1a). TReNDs were used with different linker lengths (TReND-1 – TReND-3), thus broadening the scope of possible reactivity pairs of cysteine mutants and TReNDs (Fig. 1b). Using this approach, BiP can be trapped by the AMP transfer reaction, which then will yield the covalently linked complex, FICD$^{TReND}$-BiP (Fig. 1a).

If not stated differently, the FICD construct aa 102–458 was used for all experiments as this version yielded a homogenous protein. Various cysteine substitutions (H319C$_{FICD}$, L403C$_{FICD}$,

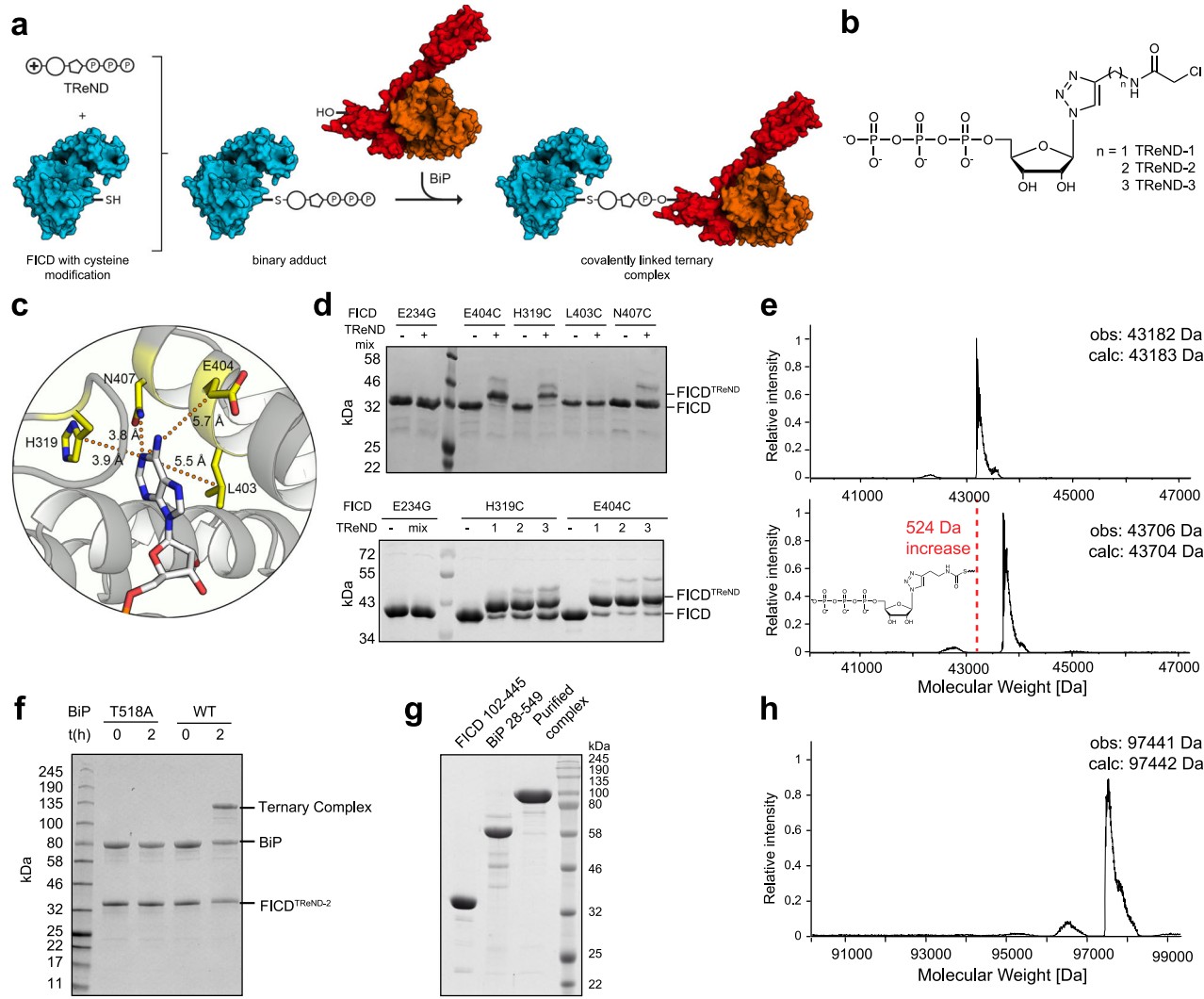

**Fig. 1 Binary adduct formation and complex formation. a** Schematic representation of covalently tethering FICD to BiP via TReNDs. First, the binary adduct is produced with TReNDs. Subsequently, a covalent ternary complex is formed with BiP in an AMPylation reaction. The ribose is displayed as pentagon, the triazole as circle, and the electrophilic moiety as encircled "+." **b** Thiol-reactive nucleotide derivatives (TReNDs) that are used in this study. **c** Selection of residues (yellow) within FICD suitable for cysteine replacement (based on the structure of FICD E234G:ATP, PDB: 4U07[23]). **d** Reactivity of TReNDs toward FICD 102–458 E234G bearing cysteine substitutions resolved by Phos-Tag™ SDS-PAGE. TReND-mix consists of TReND-1, TReND-2, and TReND-3 at equimolar concentrations. Representative gels are shown from three independent experiments. **e** Intact mass spectrometry indicating successful reaction of TReND-2 with FICD 102–458 E234G E404C. The mass deviation of unreacted FICD and FICD[TReND-2] is −1 and +2 Da, respectively. Intact mass spectrometric data for all binary adducts of FICD 102–458 E234G E404C is shown in Supplementary Fig. 1. **f** SDS-PAGE displaying the formation of a covalently linked ternary complex of FICD 102–445 E234G E404C[TReND-2] and BiP 19–654. A representative gel is shown from three independent experiments. **g** SDS-PAGE of the purified ternary complex (using BiP 28–549 T229A and FICD 102–445 T168A T183A E234G L258D E404C) submitted for crystallography. The covalent complex was purified at least two times with similar purity. **h** Intact mass spectrometry of the purified ternary complex that was used for crystallography. The mass deviation of the ternary complex is −1 Da. Source data are provided as a Source data file.

E404C$_{FICD}$, N407C$_{FICD}$) within the ATP-binding pocket of the constitutively active FICD 102–458 E234G were tested for their chemical coupling with TReNDs (Fig. 1c). The reactions were conducted in the absence of magnesium ions to prevent the hydrolysis of the binary adduct (Supplementary Fig. 1). Adduct formation was monitored by intact mass spectrometry (MS) and via Phos-Tag™ sodium dodecyl sulfate polyacrylamide gel electrophoresis (SDS-PAGE) (Fig. 1d, e and Supplementary Fig. 1). The presence of phosphates in the TReND-modified FICD increases retention in SDS-PAGE caused by Phos-Tag™ chelation. Among the cysteine substitutions tested, E404C$_{FICD}$ produced the highest yields regarding binary adduct formation, particularly upon reaction with TReND-2 and TReND-3 (binary adduct referred to as FICD[TReND]) (Fig. 1d). Additionally, we

confirmed the regioselectivity of this reaction since FICD E234G without an engineered cysteine does not react with any TReNDs although it contains one endogenous cysteine (Fig. 1e and Supplementary Fig. 1).

We then tested whether FICD 102–445 E234G E404C[TReND-2] forms a covalently linked complex upon BiP addition in the presence of Mg$^{2+}$. If not stated differently, BiP was employed as near full-length version aa 19–654 for all experiments, lacking only the N-terminal signal peptide. Indeed, an additional high molecular weight band of approximately 130 kDa appeared in SDS-PAGE indicating a covalent ternary complex between FICD and BiP that is linked via TReND-2 (Fig. 1f). Importantly, ternary complex formation proved dependent on T518$_{BiP}$, since its mutation to alanine abolished ternary complex formation

completely. This observation and MS analysis of AMPylated BiP supports $T518_{BiP}$ as the major modification site within BiP as reported before (Supplementary Fig. 2)[19,25]. The arising high molecular weight band was confirmed as the covalently linked complex via detection of their affinity tags by western blotting ($His_6$ (BiP) or Strep (FICD) tags; Supplementary Fig. 1).

We then sought to form the covalently linked complex at a preparative scale for structural analysis. For this purpose, we chose FICD 102–445 and BiP 28–549 as crystallization constructs, since they have been previously crystallized successfully[20,23]. We additionally introduced several point mutations into FICD E234G E404C to increase the homogeneity of the sample: first, $T168_{FICD}$ and $T183_{FICD}$ were substituted by alanine as we observed these residues to be autoAMPylated (Supplementary Fig. 3). Auto-AMPylation of $T183_{FICD}$ was reported before[15]. Of note, alanine substitutions of the autoAMPylation sites slightly reduced both AMPylation and deAMPylation activity of FICD (Supplementary Fig. 3). Second, $L258D_{FICD}$ was introduced to abrogate FICD dimerization, thereby preventing the formation of complexes with different stoichiometries (e.g., BiP:FICD 2:2 and 1:2 mixtures)[23]. As for BiP 28–549, the T229A mutation was introduced to keep the chaperone in the ATP state by inhibiting the ATPase activity[39]. Finally, we were able to produce ~8 mg of the complex with high purity (>95%) as indicated by SDS-PAGE and intact MS (Fig. 1g, h).

**The crystal structure of the covalently linked FICD$^{TReND}$-BiP complex.** We determined the crystal structure of the FICD$^{TReND-2}$-BiP complex via X-ray crystallography at 2.6 Å (Fig. 2a, b and see Supplementary Table 1 for data collection and refinement statistics). The FICD construct used for crystallization comprises the two N-terminal TPR motifs that are connected to the Fic domain by a linker helix. The Fic domain itself consists of a core of four α-helices ($α_1$–$α_4$) that is enveloped by three helices from the N-terminus (the inhibitory helix $α_{inh}$ and $α_{preA}$ and $α_{preB}$) and two helices from the C-terminus ($α_{postA}$ and $α_{postB}$) (Supplementary Fig. 4)[23].

The Fic domain of FICD in the complex structure does not undergo notable structural changes upon interaction with BiP when compared to previously published structures of isolated FICD as illustrated by an overall root mean square deviation (RMSD) of 0.807 Å (aligned residues: 104–434) (Fig. 2c)[31]. In contrast, in this alignment the TPR motifs of FICD undergo a twist motion upon complexation. For instance, comparing the Cα position of M135 in isolated FICD and FICD in complex indicates a movement of 9.8 Å. Accordingly, the alignment of the isolated and complexed Fic domains yielded an even lower RMSD of 0.648 Å (aligned residues: 187–434). A domain-centered alignment of the TPR motifs (aligned residues: 104–171), however, proved virtually identical in structure to the TPR motifs of previously published FICD structures (Supplementary Fig. 4)[31].

Since the complex crystallized in the presence of AMP-PNP (a β-γ non-hydrolyzable ATP analog that prevents intrinsic ATP hydrolysis of BiP), BiP is present in its ATP-bound conformation (Fig. 2a, b). The SBD and NBD of BiP are docked to each other and the conserved hydrophobic linker is tucked into the NBD of BiP. Indeed, with an overall RMSD of 0.636 Å, the complexed BiP is virtually identical to the structure of AMPylated BiP, which also features a domain-docked conformation (Fig. 2d)[20].

The complex structure reveals two distinct interfaces (Fig. 2a, b). The first interface is represented by the interaction of FICD's N-terminal TPR motif with the NBD and the conserved hydrophobic linker of BiP and buries a solvent-exposed area of 706 Å$^2$ (as determined by PISA[40]). The second interface with a size of 298 Å$^2$ comprises the interaction of the Fic domain with the SBD of BiP.

At the center of the interface between the TPR motif and the NBD, the linker residues $V415_{BiP}$ and $L417_{BiP}$ are engaged in a hydrophobic cluster together with $V241_{BiP}$ and $L128_{FICD}$ (Fig. 2e, f). Additionally, $D413_{BiP}$ as part of the conserved linker interacts with $N111_{FICD}$ and $Q112_{FICD}$ via hydrogen bonds. The hydrophobic cluster is enveloped by various residues engaged in polar interactions, such as the ionic bond of $E105_{FICD}$ and $R197_{BiP}$ and a distinct triple lysine hub consisting of $K121_{FICD}$-$D238_{BiP}$, $K124_{FICD}$-$E217_{BiP}$, $K124_{FICD}$-$D238_{BiP}$, and $K127_{FICD}$-$E243_{BiP}$. In addition, $P444_{BiP}$ from the SBD interacts with $K134_{FICD}$ and $M135_{FICD}$. Taken together, the TPR motif of FICD almost exclusively recognizes the NBD and the hydrophobic linker of BiP.

While the TPR motif binds to the conserved linker and the NBD, interestingly only few specific aa interactions for the interface of the Fic domain with the SBD were observed (Supplementary Fig. 4). Noteworthy, comparing complexed FICD with isolated FICD L258D:ATP (PDB 6I7K[31]), the α-phosphate of TReND-2 is positioned 4.9 Å from the α-phosphate of ATP[27,31]. ATP bound to isolated FICD L258D represents the AMPylation competent position, in which the α-phosphate is near the catalytic histidine ($H363_{FICD}$) and coordinates $Mg^{2+}$ with D367 of the Fic-motif. (Supplementary Fig. 4). This suggests that the determined complex structure may represent a post-catalytic state with the SBD having dissociated from the Fic domain. In support of this view, an inorganic phosphate from the crystallization buffer occupies the position of the ATP's α-phosphate and thus demonstrates that the TReND-2 has left the catalytic center in this complex.

**Confirmation of the binding interfaces of the FICD$^{TReND}$-BiP complex.** In order to assess the validity of the determined interface, we set out to identify key residues within FICD and BiP that are essential for complex formation (Fig. 3a, b). We validated these interactions by alanine substitutions of single aas and monitored the AMPylation activity of FICD L258D to AMPylate BiP via western blotting. Since FICD L258D is also able to deAMPylate BiP, we assured that within the observed time period deAMPylation was negligible and that the introduced alanine substitutions did not enhance deAMPylation activity (Supplementary Fig. 5). Most alanine substitutions exhibited a significant effect on BiP AMPylation (Fig. 3a, b), with the triple lysine hub ($K121_{FICD}$, $K124_{FICD}$, $K127_{FICD}$) being most sensitive as indicated by a reduction by >98% in each FICD mutant. $E105A_{FICD}$ reduced AMPylation by approximately 80%. Since we observed that $Y172_{FICD}$ from the neighboring asymmetric unit is involved in crystal contacts with BiP by interacting with both $R297_{BiP}$ and AMP-PNP, we wondered whether its mutation to alanine would affect AMPylation activity (Supplementary Fig. 6). However, BiP AMPylation by FICD was not significantly altered upon Y172A mutation, suggesting that this contact is of no relevance for FICD-mediated AMPylation (Fig. 3a).

In addition to verifying the importance of the interface residues of FICD, we also tested the contributions of BiP residues for AMPylation. The significance of the hydrophobic cluster was probed by alanine substitution of $V241_{BiP}$ and the two linker residues $V415_{BiP}$ and $L417_{BiP}$ that are not involved in the allosteric control of Hsp70s[41]. Indeed, we found that the substitution of $V241_{BiP}$ decreased AMPylation by >95% and substitution of V415 by 70%. No AMPylation of BiP L417A was observed further corroborating the relevance of the hydrophobic cluster for FICD–BiP interaction. Furthermore, we confirmed the contribution of $R197_{BiP}$, $E217_{BiP}$, and $D238_{BiP}$ to TPR–NBD interface, since their mutation to alanine decreased AMPylation levels by >80%. In order to exclude that the observed changes in

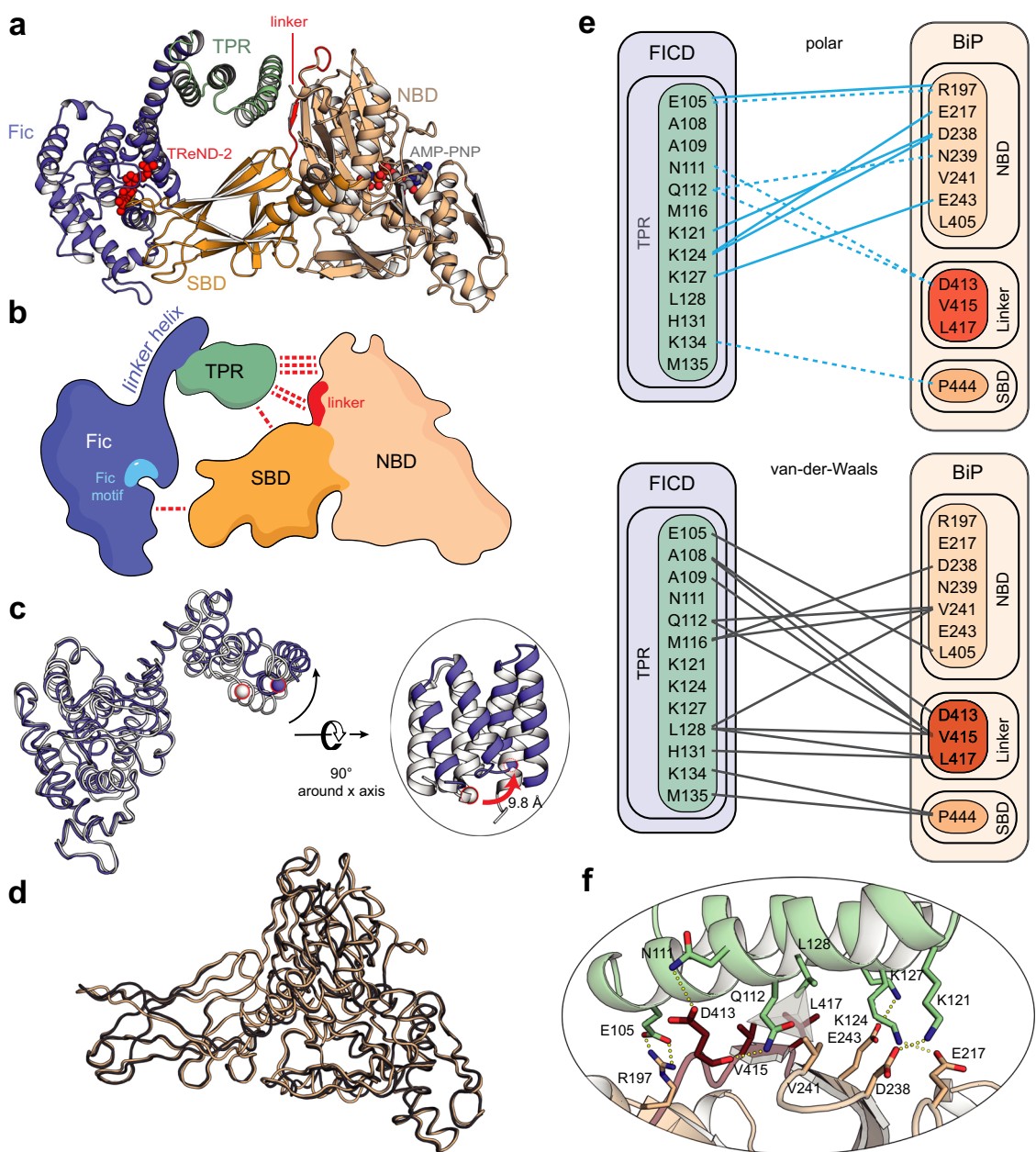

**Fig. 2 Structure of the covalent complex of FICD and BiP. a** Crystal structure of the covalently linked complex of FICD and BiP solved at 2.6 Å. FICD is represented in purple (Fic domain) and green (TPR motifs). BiP is colored in wheat (nucleotide-binding domain (NBD)) and orange (substrate-binding domain (SBD)). The conserved hydrophobic linker is highlighted in red. AMP-PNP is highlighted in light gray; TReND-2 is displayed in red. **b** Schematic overview of the complex. Colors are coded as in **a**. The Fic motif is highlighted in light blue and red dashed lines indicate specific interactions. **c** Superimposition of complexed FICD (purple) with the isolated structure of FICD (white; PDB: 6I7G[31]). Residues 104-434 were aligned. The movement of the TPR motifs is highlighted separately. The distance of Cα of M135$_{FICD}$ (red contoured spheres) from the complex and isolated FICD is calculated to be 9.8 Å. **d** Superimposition of BiP from the complex structure (wheat) with the isolated structure of BiP$^{AMP}$ (gray; PDB: 6EOF[20]). **e** Schematic representation of all polar (top) and van der Waals (bottom) contacts between the TPR motif of FICD and BiP. Polar interactions are indicated in blue with solid lines representing salt bridges and dashed lines representing hydrogen bonds. Van der Waals contacts are displayed in gray. A conclusive representation of all identified specific interactions between FICD and BiP is deposited in Supplementary Fig. 4. **f** Details of the molecular interactions of the N-terminal TPR motif of FICD with BiP. Interacting residues are highlighted and polar contacts are indicated as dashed lines in yellow. The central hydrophobic cluster is indicated by the gray pyramid. Colors are defined as in **a**.

AMPylation result from structural alterations, we determined the melting temperatures by nano dynamic scanning fluorimetry (NanoDSF), the steady-state ATPase kinetics using an enzyme-coupled assay, and potential structural changes by circular dichroism (CD) spectroscopy for each BiP mutant (Supplementary Fig. 7). All BiP mutants exhibited properties similar to BiP WT with two exceptions: BiP E217A shows an increased melting temperature of the NBD[42] together with reduced ATPase activity; BiP R197A has accelerated ATP hydrolysis as reported previously[43]. Since both E105A$_{FICD}$ and R197A$_{BiP}$ resulted in reduced BiP AMPylation to a similar extent (~80%), it is likely that the loss of the ionic bond of E105$_{FICD}$–R197$_{BiP}$ is mainly responsible for the observed AMPylation change rather than the reported effects of R197A$_{BiP}$ on ATP hydrolysis and NBD–SBD

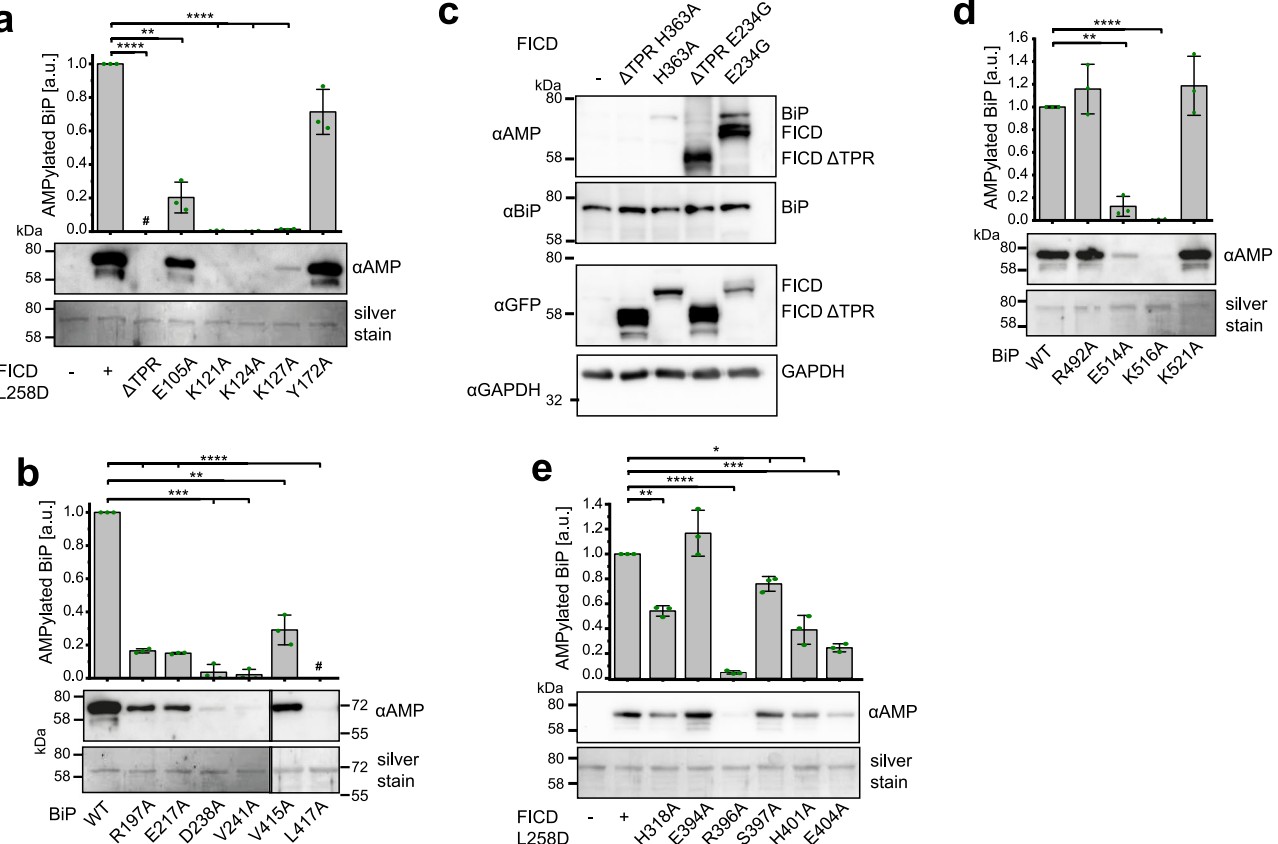

**Fig. 3 Confirmation of the TPR–NBD interface of the FICD$^{TReND}$-BiP complex and biochemical elucidation of Fic domain–SBD interaction. a** AMPylation of BiP 19–654 WT by FICD 102–458 L258D bearing distinct alanine substitutions within the TPR motif. The reaction was stopped after 30 min and the extent of AMPylation quantified via western blot with an αAMP-antibody. The experiment was performed in three independent replicates. Data are presented as mean values +/− standard deviation. Compared to FICD L258D, the $p$ values of ΔTPR, E105A, K121A, K124A, K127A, and Y172A are <0.0001, 0.0044, <0.0001, <0.0001, <0.0001, and 0.0661, respectively. **b** FICD 102–458 L258D mediated AMPylation of BiP 19–654 bearing single-point mutations within the NBD. Experimental set-up as in **a**. The experiment was performed in three independent replicates. Data are presented as mean values +/− standard deviation. Compared to BiP WT, the $p$ values of R197A, E217A, D238A, V241A, V415A, and L417A are <0.0001, <0.0001, 0.0008, 0.0003, 0.0053, and <0.0001, respectively. **c** Validation of the contribution of the TPR motif to BiP AMPylation in HEK293 cells. The lysates of HEK293 cells overexpressing different constructs of GFP-tagged FICD with (102–458) or without TPR motifs (ΔTPR: 187–458) were analyzed via western blotting. GFP was detected to verify expression, whereas GAPDH detection served as a loading control. Representative blots are shown from three biological replicates. **d** FICD 102–458 L258D mediated AMPylation of BiP 19–654 bearing single-point mutations within the SBD. Experimental conditions as in **a**. The experiment was performed in three independent replicates. Data are presented as mean values +/− standard deviation. Compared to BiP WT, the $p$ values of R492A, E514A, K516A, and K521A are 0.3374, 0.0033, <0.0001, and 0.3407, respectively. **e** AMPylation of BiP 19–654 WT by FICD 102–458 L258D bearing distinct alanine substitutions within the Fic domain. Experimental conditions as in **a**. The experiment was performed in three independent replicates. Data are presented as mean values +/− standard deviation. Compared to FICD L258D, the $p$ values of H318A, E394A, R396A, S397A, H401A, and E404A are 0.0028, 0.2580, <0.0001, 0.0197, 0.0119, and 0.0006, respectively. The number sign (#) indicates missing values due to inactivity. The significance in **a**, **b**, **d**, and **e** was determined via paired two-tailed $t$ test: *$p < 0.05$, **$p < 0.01$, ***$p < 0.001$, and ****$p < 0.0001$. a.u. arbitrary units. Source data are provided as a Source data file.

interaction[43]. Noteworthy, FICD E234G lacking the TPR motifs (ΔTPR) was devoid of AMPylation activity, highlighting the general importance of the TPR motifs for BiP binding (Supplementary Fig. 5). An observation that is supported by data obtained by size-exclusion chromatography which suggests that complex formation is dependent on the TPR motifs (Supplementary Fig. 8).

In addition, we performed a sequence alignment of FICD and BiP with their homologs from different model organisms. The alignment illustrates that the interacting residues are conserved and thus the mode of interaction is likely to be conserved too (Supplementary Fig. 9). This view is supported by AMPylation of the related chaperones human Hsp70 and Hsc70 by FICD in vitro (Supplementary Fig. 9). Furthermore, by testing FICD WT and the corresponding alanine substitutions in

deAMPylation assays, we observed that deAMPylation is indeed dependent on the TPR motif and its specific interactions that before proved important for BiP AMPylation (Supplementary Fig. 5).

To verify the contribution of the TPR motifs to BiP AMPylation within living cells, we overexpressed FICD 102–458 E234G and FICD 187–458 E234G (ΔTPR) in HEK293 cells and monitored BiP AMPylation via western blotting (Fig. 3c). Indeed, BiP AMPylation was enhanced only upon overexpression of FICD 102–458 E234G and not of FICD ΔTPR E234G, despite stronger expression of FICD ΔTPR E234G. In conclusion, the mutational analysis confirms the interactions of the TPR motifs in FICD with the NBD and conserved linker in BiP that are observed in the FICD:BiP complex crystal structure.

**Molecular dynamics (MD) simulation of the interaction of SBD and Fic domain**. The interface between the SBD of BiP and the Fic domain of FICD displayed in the crystal structure shows only few interactions and therefore does not suggest a strong contribution to the complex formation (Supplementary Fig. 4). To further evaluate the role of Fic domain in SBD binding, we combined biochemical experiments with MD simulations. In order to guide the MD simulation, we first determined the influence of selected BiP SBD alanine substitutions on FICD-mediated AMPylation (Fig. 3d). We focused on charged aas that are located at the surface of the SBD and across the Fic domain, since these positions may interact with FICD in solution even though such contacts were not detected by crystallography. The BiP substitutions $E514A_{BiP}$ in beta-sheet 7 and $K516A_{BiP}$ in loop 7,8 ($L_{7,8}$) impaired AMPylation by approximately 88% and 99%, respectively, whereas $K521A_{BiP}$ (in $L_{7,8}$) and $R492A_{BiP}$ (in $L_{5,6}$) did not notably affect AMPylation. Of note, the melting temperatures of the SBD were clearly reduced upon substitution of $E514_{BiP}$ and $K516_{BiP}$ by alanine (Supplementary Fig. 7). Hence, their observed contribution to BiP AMPylation may be partially due to structural perturbation of the SBD.

For the MD-simulation, we used the complex structure as a starting point and replaced TReND-2 by $Mg^{2+}$:ATP and $C404_{FICD}$ by the original glutamate in silico. During the MD-simulation, several interactions between the SBD and the Fic domain were suggested (Supplementary Fig. 10). We further analyzed the suggested contacts by alanine substitutions within FICD (Fig. 3e). $R396A_{FICD}$ showed a substantial decrease by 95% in the AMPylation reaction. Additionally, mutation of $H318_{FICD}$, $H401_{FICD}$, and $E404_{FICD}$ to alanine impaired BiP AMPylation by 45%, 60%, and 75%, respectively. AMPylation activity of FICD L258D S397A was only slightly reduced, whereas the activity of FICD L258D E394A was not significantly altered. Interestingly, the deAMPylation reaction by FICD WT was only mildly affected by all tested alanine substitutions (Supplementary Fig. 5). It has to be considered, however, that $R396A_{FICD}$, which in contrast to the deAMPylation reaction proved very important for the AMPylation reaction, resulted in inhomogeneous enzyme preparation as judged by size-exclusion chromatography during protein purification. The MD simulation confirmed the weak contribution of $R492_{BiP}$ to the complex formation as demonstrated by biochemical evidence (Fig. 3d and Supplementary Fig. 10). Furthermore, it suggested reasonable interaction partners for $R396_{FICD}$, $S397_{FICD}$, and $Q496_{BiP}$ yet did not clarify the molecular basis of the vital role of $K516_{BiP}$ (Supplementary Fig. 10). Overall, both the MD simulations and biochemical data confirm the relative orientation of the Fic domain and SBD to each other and agrees with the suggested post-catalytic state of the crystallized complex.

**The TPR motifs mediate specificity toward the ATP-bound state of BiP**. It was previously reported that FICD AMPylates the ATP-bound state of BiP[19]. We confirmed and extended this finding by monitoring the kinetics of ternary complex formation between BiP and $FICD^{TReND-2}$ (102–445 E234G E404C) in the presence of ADP, ATP, or no nucleotide (Fig. 4a and Supplementary Fig. 11). While ATP serves as a co-substrate for the FICD-mediated AMPylation reaction, it also binds to BiP, and thus interferes with the selective production of BiP:ADP for AMPylation. However, using the binary $FICD^{TReND-2}$ adduct that does not bind to ATP, we were able to capture BiP covalently in the distinct nucleotide states and quantify the preference of FICD toward ATP-bound BiP over ADP-bound BiP. Our data corroborate the preference of FICD for ATP-bound BiP as indicated by approximately 15× faster ternary complex formation in the presence of ATP compared to ADP. Interestingly, nucleotide-free

BiP was the worst substrate for FICD-mediated AMPylation (173× slower compared to BiP:ATP).

The conformational equilibrium of AMPylated BiP ($BiP^{AMP}$) is shifted toward the ATP-bound state as nucleotide-free BiP adopts the domain-docked conformation[20]. We wondered whether binding of ADP/ATP to $BiP^{AMP}$ still effects conformational changes within $BiP^{AMP}$, altering its substrate properties with respect to FICD-mediated deAMPylation. Indeed, $BiP^{AMP}$ is most efficiently deAMPylated when bound to AMP-PNP. In contrast to BiP AMPylation, nucleotide-free $BiP^{AMP}$ was also deAMPylated, underlining the conformational shift toward the domain-docked conformation upon AMPylation (Fig. 4b)[20]. In addition, these results indicate that $BiP^{AMP}$ is still responsive toward the bound nucleotide.

**AMPylation and deAMPylation of BiP are not directly regulated by unfolded protein substrates**. Since the substrate-binding cleft of BiP is wide open and accessible in the complex structure, we wondered whether AMPylation or deAMPylation may be influenced by binding of BiP to unfolded proteins (Supplementary Fig. 12). To test this hypothesis, we chose a well-characterized BiP substrate, the antibody domain $C_H1$ that is intrinsically unfolded[44,45]. In single-molecule Förster resonance energy transfer experiments, BiP:ADP was shown to obtain the ATP-like domain-docked conformation upon $C_H1$ binding[10]. BiP bound to $C_H1$ may therefore represent a good substrate for FICD-mediated AMPylation or deAMPylation even in the ADP-bound state. We hence tested the effect of $C_H1$ on BiP AMPylation and deAMPylation in the presence of ADP and ATP. We did not observe any major stimulating or inhibitory effect on BiP AMPylation upon preincubation of BiP:ATP with $C_H1$, likely due to the low affinity of BiP:ATP to protein substrates (Fig. 4c). To assess a possible effect of $C_H1$ on the AMPylation of BiP:ADP, we monitored the influence of $C_H1$ on covalent ternary complex formation of FICD 102–445 E234G E404C and BiP, in which the use of TReND permits the analysis of "AMPylation" of BiP:ADP. However, addition of $C_H1$ did not accelerate ternary complex formation, suggesting that BiP:ADP bound to $C_H1$ is not a favored AMPylation substrate despite obtaining the domain-docked conformation[10]. The reported $K_D$ of 7.4 µM suggests that about 75% of BiP are being bound to $C_H1$ under the chosen experimental conditions (BiP and $C_H1$ were used at a ratio of 10 µM:30 µM)[10,44]. To test a potential influence of $C_H1$ on BiP deAMPylation, $BiP^{AMP}$ was preincubated with $C_H1$ and deAMPylated in the presence of AMP-PNP or ADP. The reaction progress was monitored via western blotting (Fig. 4d). Similar to the AMPylation reaction, no significant effect of $C_H1$ was observed on the deAMPylation of $BiP^{AMP}$ bound to ADP or AMP-PNP, suggesting that neither AMPylation nor deAMPylation of BiP is directly regulated by the presence of unfolded proteins, regardless of the bound nucleotide. Of note, we observed that the ability of AMPylated BiP to bind unfolded protein substrates is impaired compared to unmodified BiP (Supplementary Fig. 12). Together, it seems unlikely that the described minor effects of $C_H1$ on BiP AMPylation and deAMPylation are physiologically relevant.

**The TPR motifs are required for AMPylation of EEF1A2**. Since the TPR motifs are essential for BiP AMPylation, we wondered whether their presence is required for FICD-mediated AMPylation of other reported substrates. We and others identified the EEF1A2 and the UMPS as putative FICD substrates[24–26]. Because EEF1A2 and UMPS share no structure and sequence similarities with BiP, we sought to uncover whether the TPR motifs of FICD contribute to their recognition. To that end, we tested the

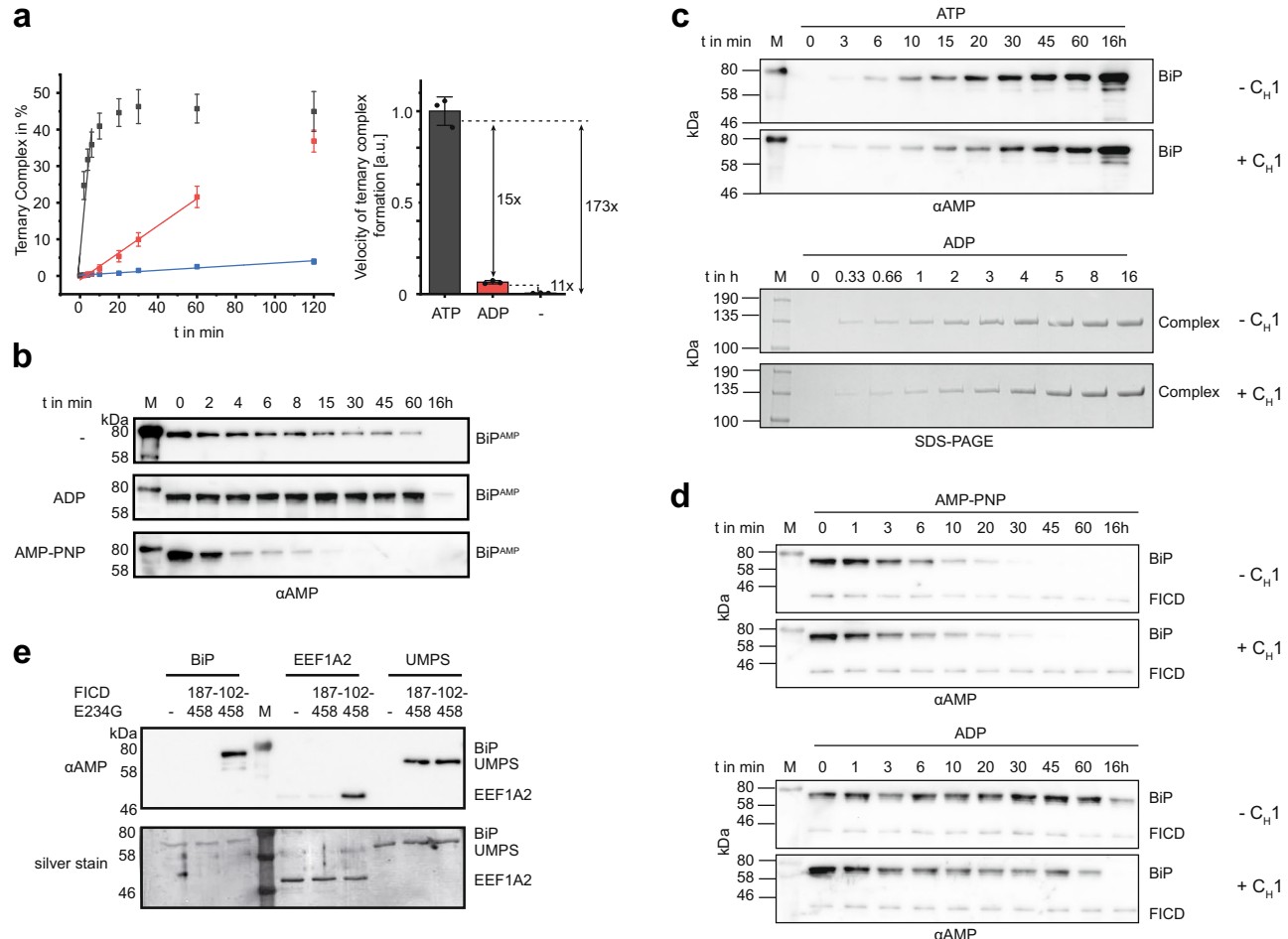

**Fig. 4 Effect of nucleotides, unfolded proteins, and TPR motifs on FICD-mediated AMPylation and deAMPylation. a** Ternary complex formation of FICD$^{TReND-2}$ (102–445 E234G E404C) and BiP 19–654 as readout for the AMPylation of BiP in the presence of ATP (gray) and ADP (red) (both at 0.5 mM) or the absence of any nucleotide (−, blue). The reaction progress was monitored via SDS-PAGE and quantified by densitometric analysis. The experiment was performed in three independent replicates. Data are presented as mean values +/− standard deviation The bar chart represents the relative mean value +/− standard deviation of the linear slope. Full gels are shown in Supplementary Fig. 11. **b** FICD 102–458 WT mediated deAMPylation of BiP$^{AMP}$ 19–654 in the presence of ATP and ADP (both at 1 mM) or the absence of any nucleotide (−) was monitored via western blotting with an αAMP antibody. **c** AMPylation of BiP with (+C$_H$1) and without (−C$_H$1) the unfolded protein substrate C$_H$1. AMPylation of BiP in the ATP-bound state of BiP was monitored in an AMPylation assay and analyzed via western blot using the αAMP antibody (upper panel). AMPylation of BiP 19–654 WT in its ADP-bound state was monitored via ternary complex formation using FICD 102–445 E234G E404C (lower panel). The full SDS-PAGE gels and the silver-stained western blots are deposited in Supplementary Fig. 13. **d** DeAMPylation of BiP was performed with purified BiP$^{AMP}$ in a deAMPylation assay in the presence of 200 μM AMP-PNP (upper panel) or ADP (lower panel). DeAMPylation of 10 μM BiP WT was performed using 0.1 μM FICD 102–458 WT (in the presence of AMP-PNP) and 2 μM FICD 102–458 WT (in the presence of ADP). The reaction progress with (+ C$_H$1) and without the unfolded protein substrate C$_H$1 (−C$_H$1) was monitored via western blot with the αAMP antibody. The silver-stained western blots are deposited in Supplementary Fig. 13. **e** Western blot with αAMP antibodies displaying AMPylation of BiP WT, EEF1A2, and MBP-tagged UMPS 1-204 by FICD 102–458 E234G and FICD 187–458 (ΔTPR) E234G. Substrates at a final concentration of 1.5 μM were incubated with 0.1 μM of FICD at 23 °C for 16 h. All experiments were performed three times independently with similar results. Source data are provided as a Source data file. (a.u.: arbitrary units).

AMPylation activity of FICD E234G ± TPR on EEF1A2 and UMPS by western blotting. Interestingly, our results suggest that FICD-mediated AMPylation of EEF1A2 requires the presence of the TPR motifs, whereas UMPS is AMPylated by FICD E234G regardless of the presence of the TPR motifs, suggesting different binding modes to these reported targets (Fig. 4e).

## Discussion
Here, we have structurally and biochemically characterized the complex of the AMP-transferase FICD and its physiological target, the ER-resident Hsp70 chaperone BiP, by covalently stabilizing their interaction using TReNDs[26]. We validated the contribution of FICD's TPR motifs to the recognition of BiP and

thus provide an explanation for FICD preferentially targeting the domain-docked conformation of BiP[19]. Moreover, we demonstrated that FICD's TPR motifs are required for AMPylation of the previously suggested target EEF1A2 but not for UMPS. Extending the FICD:BiP interface observed in the crystal structure, biochemical experiments and MD simulations led to the identification of additional key residues for target recognition both within the Fic domain of FICD and the SBD of BiP. Additionally, the presence of unfolded proteins does not directly regulate AMPylation or deAMPylation.

The importance of FICD's TPR motifs for BiP AMPylation is reminiscent of the vital contribution of the arm domain of *Histophilus somni* Fic enzyme IbpA to the AMPylation of Cdc42 and of the ankyrin repeat domain of *Legionella pneumophila* Fic

enzyme AnkX to Rab1b phosphocholination[36,37,46]. Together with the previous findings, our data suggest that the Fic domain represents the catalytic center for co-substrate transfer rather than contributing to target recognition, whereas adjacent domains that are commonly found within Fic enzymes mediate target specificity and affinity[47]. More specifically, this hypothesis is supported by the FICD:BiP complex structure in which the Fic domain appears to be captured in the process of dissociating from the SBD of BiP, while the TPR-NBD interface is still intact. Similar to the complexed Fic domain of FICD, the structure of the Fic domain of apo-IbpA is virtually identical to IbpA bound to Cdc42[36]. As for FICD's TPR motifs, most structural changes correspond to IbpA's arm domain upon complexation[36]. Interestingly, superimposition of the Fic motif in IbpA:Cdc42 and FICD:BiP reveals that the adjacent arm and TPR domain of IbpA and FICD differ in their relative position to the Fic domain (Supplementary Fig. 14). In contrast to the TPR interface in FICD:BiP, the interaction of the arm domain of IbpA with Cdc42 relies on an extensive hydrophobic interface. Our crystal structure and MD simulation studies do not confirm the role of the β-hairpin flap in recognition of the aa stretch close to the AMPylation site[36]. However, we identified $H318_{FICD}$ within the β-hairpin flap, which appears to be relevant for AMPylation. Importantly, FICD is anchored to the ER membrane via its transmembrane domain[15,48]. A stretch of >50 partially unstructured aas (JPred[49]) between the transmembrane domain and the BiP-engaging TPR motifs would allow the herein proposed mode of interaction of FICD and BiP (Supplementary Fig. 15).

While AMPylation of BiP has been reported to occur on T166/T366[14,15] or T518[19,25], our results are in favor of T518 as the only relevant modification site, as BiP T518A does not lead to ternary complex formation and only peptides comprising T518 are found to be AMPylated in liquid chromatography (LC)-MS/MS experiments. Furthermore, the covalent linker localizes to T518 in the complex structure and the therein observed contribution of the TPR motifs to BiP binding seems unlikely in the proposed AMPylation of T166/T366.

Moreover, the residues involved in the interface of the TPR motif and NBD are conserved among different species, suggesting that the mode of interaction and the targeted residue $T518_{BiP}$ is conserved too. While the contribution of the TPR–NBD interface proved important for both BiP AMPylation and deAMPylation, the interface of the Fic domain and SBD seems to have different qualities for either reaction as specific alanine substitutions affected the two reactions to a different extent.

Recent structural and biochemical studies suggested that monomeric FICD is AMPylation active in contrast to dimeric FICD (AMPylation inactive, which acts as a deAMPylase)[29,31,32]. While the positioning of phosphates of the bound ATP as well as the side chain of E234 in the structures of monomeric and dimeric FICD differ, the Fic domain does not undergo notable structural changes[31]. Since the structure of complexed FICD bears a E234G mutation, no conclusion can be drawn on the side chain's positioning. The Fic domain of complexed FICD is virtually identical to the Fic domain of dimeric and monomeric FICD structures, indicating that the proposed conformational plasticity of the Fic domain that governs the positioning of E234 side chain is difficult to grasp via crystallography[23,31]. Interestingly, the TPR motifs of monomeric FICD in complex with ATP and AMP-PNP were shown to be flipped by 180 degrees[31]. The complex structure of FICD and BiP, however, shows that only a minor twist of the TPR motifs and not a major reorientation contributes to BiP binding and AMPylation. Hence, the observed flip in FICD L258D:ATP is likely to be a crystallographic artifact.

BiP complexed with FICD is, similar to AMPylated BiP, in its ATP-bound state, which is the preferred state for AMPylation[19,20]. This preference is well explained by the specific recognition of the conserved hydrophobic linker inserted into the NBD by FICD's TPR motifs[19]. Of note, this interaction mode is fundamentally different from other Hsp70-associated proteins that mostly bear TPR motifs, that specifically bind to the C-terminal peptide of Hsp70 via the TPR's concave surfaces[50,51]. Interestingly, the herein described "side-on" TPR interaction interface has been described before[52], yet presents a rather rare interaction mode of TPR motifs that usually interact via their convex or concave surfaces[53]. The recognition of the conserved linker by FICD implicates that only the monomeric active pool of BiP is efficiently targeted in cells and inactivated by AMPylation[16,19]. The finding that deAMPylation is also dependent on the TPR motifs suggests a similar mode of interaction for the deAMPylation complex. The hypothetical deAMPylation complex of dimeric FICD bound to one or two BiP molecules is at least sterically conceivable (Supplementary Fig. 16).

Interestingly, we observed a strong preference of ATP-bound $BiP^{AMP}$ as a substrate for FICD-mediated deAMPylation. On the one hand, this observation is in agreement with previous biochemical evidence indicating conformational changes of monomeric $BiP^{AMP}$:ATP in comparison to monomeric $BiP^{AMP}$:ADP[20]. On the other hand, previous nuclear magnetic resonance experiments suggested irresponsiveness of $BiP^{AMP}$ to the bound nucleotide, ADP or ATP[54]. In contrast to AMPylation of nucleotide-free BiP, we observed that FICD-mediated deAMPylation of nucleotide-free BiP is possible. This is well explained by the shift of $BiP^{AMP}$ toward the domain-docked conformation[20]. In the physiological context, however, $BiP^{AMP}$ is likely to be bound to ATP, due to high cellular levels of ATP.

The $C_H1$ antibody domain as a BiP substrate does not directly regulate FICD-mediated AMPylation or deAMPylation in vitro. While $C_H1$ binding to BiP was previously shown to shift the conformational equilibrium of BiP toward the domain-docked conformation even in the presence of ADP, we did not observe any stimulation or inhibition of FICD-mediated AMPylation or deAMPylation in vitro[10]. While the shift toward the domain-docked conformation upon $C_H1$ binding is favorable for FICD-mediated BiP AMPylation, $C_H1$-binding may also lead to rearrangements within the loops or partial obstruction of loop regions within the SBD of BiP that are relevant for the interaction with FICD. We speculate that these two competing processes abrogate each other. The deAMPylation of BiP, however, does not seem to be directly regulated by unfolded proteins since its ability to bind unfolded protein substrates is impaired. Our observation is in line with previous results, demonstrating that the modified species of BiP is not complexed to protein substrates[55].

The finding that the TPR motifs of FICD are crucial for AMPylation of BiP and EEF1A2 but not for the AMPylation of UMPS may provide a means to distinguish between physiological substrates of FICD and unspecific AMPylation events. While EEF1A2 and UMPS reside in the cytosol, FICD was shown to localize within the ER under normal conditions[15,56]. However, it is currently uncertain whether FICD localization changes upon certain stressors or in different cell lines and tissues. While several studies identified EEFA1A2 and UMPS as AMPylation substrates of FICD[24–26], recent work has shown that only BiP AMPylation was dependent on the expression of FICD, in contrast to EEF1A2 and UMPS that were not enriched with an N6-propargyl ATP probe[57]. It cannot be excluded that yet unidentified cytosolic enzymes or enzyme classes might be responsible for AMPylation of EEF1A2 and UMPS. Speculatively, some pseudokinases may reside in the cytosol and possess AMPylation activity. The AMPylation activity of pseudokinases has recently been

demonstrated for the mitochondrial pseudokinase SelO[58]. Thus, even though EEF1A2 and UMPS can be AMPylated by FICD in vitro, the physiological significance of this observation remains enigmatic.

In summary, we solved the structure of the human AMP transferase FICD in complex with its substrate the Hsp70 chaperone BiP and uncovered the molecular principles underlying FICD's specificity toward ATP-bound BiP by discovering a novel mode of interaction of TPR motifs with Hsp70 chaperones.

## Methods

**Cloning of constructs**. All FICD constructs were cloned into a pSF421 vector with an N-terminal 6xHis-GFP- or 6xHis-Halo®-tag. BiP was cloned into a pProEX vector with an N-terminal 6xHis-tag. The $C_H1$ domain from murine Mak33 was cloned without affinity tag version into pET28b. All tags were separated from the gene of interest by a tobacco-etch virus (TEV) protease cleavage site. FICD 102–458 and FICD 187–458 were subcloned into the pAC vector for transient mammalian expression under control of the CMV promoter. This construct was equipped with an N-terminal ER targeting sequence (BiP 1–19) and a C-terminal GFP-tag (with the KDEL ER retention sequence) to verify the expression. Point mutations were introduced using the Q5® Site-Directed Mutagenesis Kit (NEB) and performed as described by the manufacturer. All oligonucleotides were obtained from Integrated DNA Technologies, Inc. (IDT, Coralville, AI, USA) and are summarized in Supplementary Tables 2 and 3. The cloned constructs were sequenced by GATC/Eurofins Scientific or Microsynth. NCBI accession numbers for the proteins used in this study are: FICD 187–458 (ΔTPR), 102–445, 102–458 (NP_009007.2), human BiP 19–654 and 28–549 (NP_005338.1), EEF1A2 1–463 (NP_001949.1), UMPS 1-204 (NP_000364.1).

**Protein expression and purification**. All proteins were expressed in *E. coli* Rosetta™ (DE3) Competent Cells (Novagen). Generally, 100 ng of plasmid was transformed into *E. coli* Rosetta™ (DE3) and a single colony was picked for starting an overnight preculture at 37 °C in 10 mL LB. The preculture was directly transferred to 1 L prewarmed LB media and grown to an $OD_{600}$ of 0.5. Expression was induced with 0.5 mM IPTG. FICD and UMPS were expressed at 23 °C for 16 h and BiP and EEF1A2 were expressed at 37 °C for 3 h. Expression of the murine $C_H1$ domain was performed in BL21 at 37 °C for 16 h after induction with 1 mM IPTG at $OD_{600}$ of 0.6 as previously described[44]. After expression, cells were harvested at $7000 \times g$ for 15 min. The cell pellets were resuspended in cold phosphate-buffered saline (PBS) and subsequently centrifuged at $3000 \times g$ for 30 min. Pellets were flash-frozen in liquid nitrogen and stored at −20 °C.

Cell pellets were resuspended in ice-cold Buffer A (50 mM HEPES-NaOH pH 7.4, 500 mM NaCl, 1 mM $MgCl_2$, 1 mM β-mercaptoethanol) and homogenized by Silent Crusher M (Heidolph Instruments GmbH & CO. KG). After addition of DNAse I, cells were lysed with a Constant Cell Disruption System (Constant Systems Limited, UK) at 1.8 kbar and 1 mM PMSF protease inhibitor was added. Cell debris was separated by centrifugation at $50,000 \times g$ for 30 min before the lysates were loaded onto a $Ni^{2+}$-NTA IMAC column (Bio-Rad) using the NGC™ Liquid Chromatography System (Bio-Rad). Gradient elution of bound proteins was achieved by addition of Buffer B (50 mM HEPES-NaOH pH 7.4, 500 mM NaCl, 1 mM $MgCl_2$, 1 mM β-mercaptoethanol, 500 mM imidazole). Fractions that were of >95% purity were pooled. BiP was purified in modified Buffer A (50 mM HEPES–NaOH pH 7.4, 400 mM NaCl, 20 mM imidazole) and modified Buffer B (50 mM HEPES–NaOH pH 7.4, 400 mM NaCl, 500 mM imidazole).

When cleavage of the solubility/affinity tags was desired, proteins were dialyzed overnight at 4 °C against 20 mM HEPES–NaOH pH 7.4, 200 mM NaCl, 1 mM $MgCl_2$ and 1 mM β-mercaptoethanol in the presence of TEV protease at a ratio of 1 mg homemade TEV protease per 40 mg protein substrate. BiP was dialyzed against modified dialysis buffer (20 mM HEPES–NaOH pH 7.4, 100 mM NaCl). MBP-tagged UMPS was dialyzed without TEV against 20 mM HEPES–NaOH 100 mM NaCl, and 1 mM TCEP, concentrated, flash-frozen in liquid nitrogen, and stored at −80 °C. For other proteins, a reverse IMAC was performed after dialysis to remove both tags and TEV protease. Proteins were concentrated to 0.5–2 mL with Amicon® Ultra Centrifugal Filters (Merck-Millipore) and purified by size-exclusion chromatography using a HiLoad Superdex 16/600 75 pg (for FICD constructs) and 200 pg (for BiP constructs) Gel Filtration Column (GE-Healthcare). FICD was purified in size-exclusion buffer (20 mM HEPES-NaOH pH 7.4, 150 mM KCl, 1 mM $MgCl_2$, 1 mM TCEP, 10% Glycerol). The size-exclusion buffer for FICD constructs bearing a cysteine mutation for binary adduct formation was supplemented with 1 mM EDTA instead of 1 mM $MgCl_2$. BiP 19–654 was purified in HKM buffer (20 mM HEPES–KOH pH 7.4, 150 mM KCl, 10 mM NaCl, and 1 mM $MgCl_2$), whereas BiP 28–549 was purified in modified HKM buffer (20 mM HEPES–KOH pH 7.4, 200 mM KCl, 10 mM $MgCl_2$).

Cells overexpressing tagless murine $C_H1$ were disrupted as described above. After cell disruption and centrifugation of lysates, the supernatant was discarded and the pellet solubilized in 1% Triton X-100 at 4 °C for 16 h. The inclusion bodies were separated from the soluble fraction via centrifugation ($50,000 \times g$ for 30 min) and washed twice in 50 mM Tris/HCl pH 7.5, 5 mM EDTA, and 5 mM NaCl and

stored at −20 °C. The inclusion bodies were dissolved in 25 mM Tris pH 7.8, 5 mM EDTA, 8 M Urea, and 10 mM β-mercaptoethanol for 2 h at room temperature and solubilized proteins were separated via centrifugation ($50,000 \times g$ for 30 min). The supernatant was loaded onto an anion-exchange column (HiTrap Q HP, GE Healthcare) and $C_H1$ purified with Low salt buffer (25 mM Tris pH 7.8, 5 mM EDTA, 5 M Urea) and high salt buffer (25 mM Tris pH 7.8, 5 mM EDTA, 8 M Urea, 1 M NaCl). Refolding of the $C_H1$ domain was performed by dialysis against at least 20 volumes of 250 mM Tris pH 8.0, 5 mM EDTA, and 1 mM oxidized glutathione at a concentration of <0.1 mg/mL at 4 °C for 16 h. After refolding, $C_H1$ was concentrated and purified via size-exclusion chromatography (Superdex 16/600 75 pg) in phosphate-buffered saline. The identity of the protein was confirmed by SDS-PAGE and MS.

6xHis-EEF1A2 was purified as reported previously[24]. In brief, cells were resuspended in 50 mM Tris-HCl pH 7.5, 500 mM NaCl, 5% glycerol, and 1 mM PMSF and disrupted via sonication. After addition of DNAse I, the soluble fraction was removed by centrifugation and the pellet solubilized in 50 mM Tris–HCl pH 7.5, 500 mM NaCl, 5% glycerol, and 6 M Urea. The insoluble fraction was removed by centrifugation and EEF1A2 purified on a $Ni^{2+}$-NTA IMAC column using 50 mM Tris-HCl pH 7.5, 500 mM NaCl, 5% glycerol, and 6 M Urea as Buffer A and 50 mM Tris-HCl pH 7.5, 500 mM NaCl, 5% glycerol, 6 M Urea, and 500 mM imidazole as Buffer B. The elongation factor was refolded via dialysis against 50 mM Tris-HCl pH 7.5, 150 mM NaCl, and 10% glycerol. Pooled fractions were further concentrated, aliquoted, and flash-frozen in liquid nitrogen. Proteins were stored at −80 °C.

**Binary adduct and ternary complex formation**. FICD bearing an additional cysteine was incubated at a concentration of 50 µM with 100 µM TReND in binary adduct buffer (20 mM HEPES–KOH pH 7.4, 100 mM KCl, 1 mM EDTA) for 16–20 h at 23 °C. Ternary complex formation was achieved by incubation of 30 µM FICD binary adduct and 30 µM BiP in AMPylation buffer (20 mM HEPES–KOH pH 7.4, 100 mM KCl, 4 mM $MgCl_2$, 1 mM $CaCl_2$) for 2 h at 23 °C. For ternary complex formation in the presence of $C_H1$, 20 µM BiP 19–654 WT was pre-incubated with 50 µM $C_H1$ (+$C_H1$) or buffer (−$C_H1$) in the presence of 1 mM ADP for 16 h at 23 °C. Subsequent ternary complex formation was performed with 10 µM FICD 102–445 T168A T183A E234G L258D E404C$^{TReND-2}$ and 10 µM BiP 19–654 WT (with or without final 25 µM $C_H1$).

**Purification of FICD$^{TReND-2}$-BiP ternary complex**. The covalently linked ternary complex FICD$^{TReND-2}$-BiP was prepared as described above with purified FICD 102–445 T168A T183A E234G L258D E404C and 6xHis-BiP 28–549 T229A. For purification, the complex was loaded onto a $Ni^{2+}$-NTA IMAC column (Bio-Rad) to deplete unreacted FICD molecules and purified with regular Buffers A and B. The His-tag of covalently linked BiP was cleaved upon addition of TEV protease at a ratio of 1:40 and the complex was dialyzed against 20 mM HEPES pH 7.4, 150 mM KCl, 1 mM $MgCl_2$, and 1 mM β-mercaptoethanol at 4 °C for 16 h. After concentration and addition of 1 mM ATP, the complex was further purified on a HiLoad Superdex 26/600 200 pg Gel Filtration Column (GE Healthcare) connected in series to a HiLoad Superdex 26/600 75 pg Gel Filtration Column (GE Healthcare) in 20 mM HEPES–KOH pH 7.4, 150 mM KCl, 0.5 mM $MgCl_2$, 0.5 mM TCEP, and 100 µM ATP. Finally, the complex was dialyzed overnight at 4 °C against 20 mM HEPES–KOH pH 7.4, 150 mM KCl, 1 mM EDTA, and 1 mM dithiothreitol (DTT), concentrated to a final concentration of 6.9 mg/mL, aliquoted, and flash-frozen in liquid nitrogen.

**Crystallography**. Directly before crystallization trials, 1 mM AMP-PNP, 1 mM $MgCl_2$, and glycerol to a final concentration of 10% (v/v) were added to the complex (final concentration of 5.5 mg/mL). Sitting drop crystallization trials were carried out at 19 °C, by mixing equal volumes (100 nL) of reservoir solution and protein solution. Crystals grew in a condition containing 0.2 M NaCl, 0.1 M Na/K phosphate pH 6.2, and 10% (w/v) PEG 8000. Crystals were soaked in cryo-solutions containing the crystallization mother liquor supplemented with 25% (v/v) glycerol, mounted onto a cryoloop (Hampton Research), and immediately flash-cooled in liquid nitrogen. Diffraction data were collected at EMBL beamline P13 at the PETRA III storage ring (DESY, Hamburg, Germany). Diffraction data were processed using DIALS[59] and scaled with Aimless from the CCP4 suite[60,61].

The structure was solved by molecular replacement using the CCP4go automatic procedure[62]. Briefly, the program MoRDa[63] used search models based on structures 4U04 and 5O4P to solve the structure by molecular replacement. The initial solution was further constructed and partially refined respectively with CCP4Build[62], Buccaneer[64], and Refmac[65]. The automatically built model was then corrected and further built manually with COOT[66] and refined using the PHENIX suite[67] and the PDB_REDO web server[68]. The quality of the final model was assessed using the wwPDB validation server[69] and the Molprobity server[70].

An OMIT map where the ligand TReND-2 as well as the two crosslinked residues have been omitted was calculated using the PHENIX suite[67]. Data collection and refinement statistics are gathered in Supplementary Table 1.

The atomic coordinates and structure factors have been deposited in the PDB with accession code 6ZMD.

**AMPylation assay**. FICD and BiP were incubated at a molar ratio of 1:100. Typically, 0.1 μM of FICD E234G or FICD L258D were incubated with 10 μM BiP in the presence of 1.5 mM ATP in AMPylation buffer (20 mM HEPES–KOH pH 7.4, 100 mM KCl, 4 mM MgCl$_2$, 1 mM CaCl$_2$). In AMPylation assays with C$_H$1, BiP was preincubated with C$_H$1 for 100 min before AMPylation reactions were started by addition of FICD. Reactions were stopped after 30 min (if not stated differently) by addition of Laemmli buffer and boiling at 95 °C for 5 min.

**DeAMPylation assay**. In order to produce AMPylated BiP, 6xHis-BiP was incubated for 16 h at 23 °C with FICD 102–458 E234G at a ratio of 1:20 using the already described AMPylation reaction conditions. 6xHis-BiP was affinity purified using the same conditions as described in the section for protein purification and dialyzed twice against 20 mM HEPES–KOH pH 7.4, 150 mM KCl, and 1 mM EDTA at 4 °C. Alternatively, AMPylated BiP was produced by incubation with FICD 45-458 E234G at a molar ratio of 1:100 and dialyzed twice against 20 mM HEPES–KOH pH 7.4, 150 mM KCl, and 1 mM EDTA at 4 °C. Purified BiP$^{AMP}$ was aliquoted and flash-frozen in liquid nitrogen. For deAMPylation, FICD WT and BiP were incubated at a molar ratio of 1:50, if not indicated differently. FICD 102–458 WT (0.4 μM) was incubated with 20 μM BiP$^{AMP}$ in HKM buffer (20 mM HEPES–KOH pH 7.4, 150 mM KCl, 10 mM MgCl$_2$), if not stated differently. In all experiments with C$_H$1, BiP was preincubated with C$_H$1 for 100 min before deAMPylation reactions were started by addition of FICD. Reactions were stopped at a specific time point by addition of Laemmli buffer and boiling at 95 °C for 5 min.

**Western blots**. Protein samples were resolved via SDS-PAGE using homemade 12% or 15% acrylamide gels and transferred to Immobilon-P PVDF membranes (Merck-Millipore) for 2 h at 320 mA by a V10-SDB Semi-Dry Blotter (Scie-Plas, Cambridge, UK). Membranes were washed with TBS-T buffer and blocked with Roti®-Block (Carl Roth) for at least 1 h at 4 °C. The primary monoclonal αAMP antibody[71] was subsequently added at a 1:1000 dilution and incubated overnight at 4 °C. For Strep-tag detection, Strep-Tactin®-HRP (Iba-Lifesciences), and for His$_6$-tag detection, the SuperSignal® West HisProbe™ Kit (Thermo Scientific), was applied according to the manufacturer's instructions. The rabbit αGFP primary antibody for GFP detection (Life Technologies, #A11122) was used at 1:2000, the rabbit αGRP78 primary antibody for BiP detection (Thermo Scientific, #PA5-34941) was used at 1:5000,and the mouse αGAPDH primary antibody for GAPDH detection (Santa Cruz, #sc-47724) was used at 1:1000. After washing, either goat secondary α-mouse-HRP-antibody (#31430, Thermo Scientific) was added at a 1:20,000 dilution or goat secondary α-rabbit-HRP-antibody (#12-348, Sigma) was added at a 1:40,000 dilution and incubated at room temperature for 1 h. Membranes were washed and visualized by INTAS ECL CHEMOCAM (Software: ChemoStar Touch) (Intas Science Imaging) after incubation with WesternBright™ ECL-Spray (Advansta) for 2–3 min. Stripping of membranes was performed using Roti®Free Stripping-Buffer (Carl Roth). Membranes were reblocked after several washing steps and the next primary antibody was added. Colloidal silver staining of the membranes was performed exactly as reported previously[72]. In both AMPylation and deAMPylation assays, the reaction progress was determined semi-quantitatively by densitometric analysis (Software: Image Lab). In deAMPylation assays for the comparison of FICD (L258D) with various alanine substitutions, the relative deAMPylation activity was determined as the normalized difference of BiP$^{AMP}$ with and without (−) enzyme. Some mutants exhibited very low deAM-Pylation activity, which, due to the natural distribution of values in western blotting, in some replicates yielded slightly negative values after subtraction. These negative values were set to "0." Uncropped gels and western blots are provided in the Source data file.

**Phos-Tag™ electrophoresis**. To visualize the successful formation of binary adducts, 3 μg of protein was loaded on an acrylamide gel consisting of a 4.5% acrylamide stacking gel and a 12% acrylamide resolving gel. The resolving gel was supplemented with 25 μM Phos-Tag™ AAL-107 (NARD Institute Ltd.) and 100 μM MnCl$_2$. The gels were run at 30 mA for 60–90 min.

**Analytical high-performance LC (HPLC) experiments**. To analyze the binding of C$_H$1 to BiP, C$_H$1 was first labeled with NHS-Fluorescein (#46410, Thermo Scientific) according to the manufacturer's instructions with minor adjustments. The labeling was performed in PBS at pH 7.4 at a molar ratio of 1:1 (C$_H$1: NHS-Fluorescein) for 1 h at room temperature. Residual label was removed via desalting of C$_H$1 into HKM buffer. BiP was incubated for 16 h with C$_H$1 in HKM buffer supplemented with 1 mM ADP. Complex formation was analyzed via size-exclusion chromatography (Superdex 10/300 200 pg column) at a flow rate of 0.5 mL/min at a Shimadzu UFPLC system (Prominence Series) using the Lab solutions software (Shimadzu). The absorption was measured at 280 and 496 nm. The binding of FICD to BiP was analyzed by incubating 5 μM FICD 102–458 L258D with 100 μM BiP 19–654 T229A T518A in HKM buffer supplemented with 5 mM ATP for at least 2 h. The proteins were separated on a Superdex 10/300 75 pg using the same parameters and buffer as described above.

**Melting point determination via NanoDSF**. All BiP mutants were diluted to 1 mg/mL in HKM buffer. The samples were loaded into the standard capillaries (nanotemper, #PR-C002) and analyzed via the Prometheus NT.48 (nanotemper) (Software: PR.ThermControl) at a gradient of 1 °C/min ranging from 20 to 80 °C. The melting points were derived from the ratio of fluorescence at 350/330 nm.

**Steady-state ATPase assay**. The ATPase rate of BiP and BiP mutants was determined by applying an ATP regenerating system in which the consumption of NADH directly corresponds to the hydrolysis of ATP[73]. The absorption of NADH was measured at 340 nm in a TECAN Spark microreader (Tecan) in Corning® 384-well plates (#3640). The assay was performed at 37 °C with 1 mM ATP and 2 μM BiP.

**CD spectroscopy**. All BiP mutants were diluted to 0.1 mg/mL in 10 mM potassium phosphate pH 7.5, 150 mM KF, 1 mM MgCl$_2$. The spectra were collected at 25 °C in a 0.1 mm cuvette from 185 nm to 260 nm on a Chirascan CD spectrometer (Applied Photophysics) (Software: Chirascan Spectrometer Control Panel Application Version). The instrument was set to 0.5 nm bandwidth, 0.5 s response, and 0.5 nm data pitch. The spectra were analyzed with Pro-Data Viewer and background subtracted. Each curve represents the mean of three measurements.

**LC-MS sample preparation**. High-resolution mass spectra of binary adducts (FICD$^{TReND}$) and the covalently linked ternary complex (FICD$^{TReND-2}$-BiP) were recorded on an Agilent 6230 Series TOF mass spectrometer coupled to an Agilent 1290 Infinity II LC system (HPLC column: Agilent, Poroshell C8, 2.1 mm × 75 mm, 5 μm particle size, flow: 600 μL/min, gradient of eluent A: milliQ H$_2$O + 0.1% formic acid (FA) and eluent B: acetonitrile (ACN) + 0.1% FA, ionization method: electrospray ionization). Samples were injected as 1 μL from 0.1 to 0.5 mg/mL corresponding buffer stock solutions with time-based online desalting (0 – 1 min to waste). Ion spectra were analyzed using Agilent MassHunter Qualitative Analysis B.07.00 and spectra were deconvoluted using either maximum entropy algorithm (all binary adducts without Mg$^{2+}$ and ternary complexes) or pMod algorithm (binary adducts with Mg$^{2+}$).

High-resolution mass spectra of BiP and BiP$^{AMP}$ were recorded on a Bruker maXis II™ QTOF mass spectrometer (Software: Bruker Compass HyStar) coupled to an Bruker Elute LC system (monolithic column Thermo ProSwift RP-4H (50 mm × ID 1 mm), flow: 300 μL/min, gradient of eluent A: milliQ H$_2$O + 0.1% FA and eluent B: ACN + 0.1% FA (5% B to 80% B in 2 min), ionization method: electrospray ionization). Samples were injected as 2 μL from 0.3 mg/mL. Ion spectra were analyzed from the base peak chromatogram using Bruker Compass DataAnalysis 5.1, and spectra were deconvoluted using the maximum entropy algorithm.

**LC-MS/MS sample preparation**. Purified FICD (102–458, E234G) preincubated with 1 mM ATP for 16 h was diluted with 0.1 M triethylammonium bicarbonate (Thermo Fisher Scientific) and 1% sodium deoxycholate (SDC, Sigma), and was reduced in the presence of 10 mM DTT (Sigma Aldrich) for 30 min at 60 °C followed by cysteine alkylation with 20 mM iodo acetamide (Sigma Aldrich) for 30 min at 37 °C in the dark and enzymatic degradation with sequencing-grade trypsin (Promega) overnight at 37 °C. Digestion was quenched with 1% FA, precipitated SDC was removed by centrifugation for 5 min at 14,000 × g, and the supernatant was dried in a vacuum centrifuge.

**LC-MS/MS acquisition, data analysis, and processing**. Chromatographic separation of peptides was achieved by nano UPLC (nanoAcquity system, Waters) with a two buffer system (buffer A: 0.1% FA in water, buffer B: 0.1% FA in ACN). Attached to the UPLC was a peptide trap (180 μm × 20 mm, 100 Å pore size, 5 μm particle size, Symmetry C18, Waters) for online desalting and purification followed by a 25-cm C18 reversed-phase column (75 μm × 200 mm, 130 Å pore size, 1.7 μm particle size, Peptide BEH C18, Waters). Peptides were separated using a 60-min gradient with increasing ACN concentration from 2% to 30% ACN. The eluting peptides were analyzed on a Quadrupole Orbitrap hybrid mass spectrometer (QExactive, Thermo Fisher Scientific). Here, the ions being responsible for the 12 highest signal intensities per precursor scan (1 × 10$^6$ ions, 70,000 Resolution, 120 ms fill time) were analyzed by MS/MS (HCD at 28 normalized collision energy, 1 × 10$^5$ ions, 17,500 Resolution, 50 ms fill time) in a range of 400–1300 $m/z$. A dynamic precursor exclusion of 20 s was used.

LC-MS/MS were searched with the Sequest algorithm integrated in the Proteome Discoverer software (v 2.0, Thermo Fisher Scientific) against the protein sequence of FICD (102–458, E234G) and a contaminant data protein sequence database. Carbamidomethylation was set as fixed modification for cysteine residues and the oxidation of methionine, phosphoadenosine at serine and threonine and tyrosine, and pyro-glutamate formation at glutamine residues at the peptide N-terminus, as well as acetylation of the protein N-terminus were allowed as variable modifications. Potential peptides with AMPylation were manually inspected to confirm the presence of characteristic AMP signals at 136.0623 $m/z$ (adenine fragment), 250.0940 $m/z$ (adenosine fragment), and/or 348.0709 $m/z$ (AMP fragment) and only then accepted as a valid peptide AMPylation.

**Mammalian cell culture**. HEK293 cells (DSMZ ACC-305) were cultivated in Dulbecco modified Eagle medium (Thermo Fisher Scientific) with 10% (v/v) fetal bovine serum (Thermo Fisher Scientific). Cells were seeded in 6-well plates (Sarstedt) at a density of $3.5 \times 10^5$ per well. After 24 h, 2 μg of purified vector DNA were used to transiently transfect the cells employing Lipofectamine 2000 (Thermo Fisher Scientific) according to the manufacturer's instructions. After 24 h of expression, cells were washed twice in PBS and spun down at 4 °C and $300 \times g$ for 5 min. All following steps were performed on ice. The cell pellet was further lysed with RIPA buffer (Thermo Fisher Scientific) supplemented with cOmplete EDTA-free protease inhibitor (Roche) according to the manufacturer's instructions. Protein concentration was determined via Bradford (#5000006, Bio-Rad).

**MD simulation**. The crystal structure of the FICD$^{TReND-2}$-BiP complex served as the start structure. The linker was removed and instead the C404$_{FICD}$ was replaced by glutamate. ATP and its attached Mg ion were taken from the ATP-bound FICD L258D (PDB 6I7K[31]) structure. AMBER ff14SB force-field parameters[74] for proteins and ATP parameters from the AMBER parameter database[75] were used for the simulation. The complex was inserted in an octahedron box with a minimum distance of 8 Å from the edge, filled with water molecules. The TIP3P water model was employed to explicitly model the solvent interactions. Sodium and chloride ions were inserted into the box to reach a salt concentration of 0.1 M. Protein atoms were restrained during the NVT and NPT ensemble simulations. The solvated box was energy minimized for 5000 steps. Then a heating (25 ps) and a density equilibration (50 ps) followed to reach the temperature 310 K. Finally, an equilibration simulation of 150 ps in NPT simulation was performed. During the equilibration phase, the distance between the oxygen atom connecting the ATP sugar base to the α-phosphate and the side chain of the H363$_{FICD}$ residue were restrained to their initial distance using a harmonic potential with a decreasing force constant starting at 20.0 kcal/mol/Å$^2$. Production simulations were performed without any restraints. The pmemd version of the AMBER 18 software package[76] was used employing hydrogen mass repartitioning feature of Parmed tool, which allows a simulation time step of 4 fs. Long range interactions were included using the particle mesh Ewald method combined with periodic boundary conditions and an 8.5-Å cut-off for real-space non-bonded interactions. Trajectories were processed and analyzed using CPPTRAJ program[77].

**Reporting summary**. Further information on research design is available in the Nature Research Reporting Summary linked to this article.

## Data availability

Atomic coordinates and structure factors of the FICD$^{TReND}$-BiP complex have been deposited in the Protein Data Bank with accession code 6ZMD. The mass spectrometric proteomics data have been deposited to the ProteomeXchange Consortium via the PRIDE partner repository with the dataset identifier PXD022869. The AMBER force-field parameters can be found in the AMBER parameter database (http://amber.manchester.ac.uk). The data that support the findings of this study are available from the corresponding author upon reasonable request. Source data are provided with this paper.

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

## Acknowledgements

The synchrotron MX data were collected at beamline P13 operated by EMBL Hamburg at the PETRA III storage ring (DESY, Hamburg, Germany). We would like to thank Guillaume Pompidor for the assistance in using the beamline. We also acknowledge technical support from the SPC facility at EMBL Hamburg. We thank Johannes Buchner at TU Munich for sharing the plasmids of BiP and $C_H1$ and providing Hsp70 and Hsc70. This work was performed within the framework of SFB 1035 (German Research Foundation DFG, Sonderforschungsbereich 1035, Projektnummer 201302640, projects B02 and B05). Mass spectrometry was funded by the Deutsche Forschungsgemeinschaft (DFG, German Research Foundation) – Projektnummer INST 152/859-1 FUGG. C.H. thanks Knut and Alice Wallenberg Foundation (Sweden) and the Swedish research council (grants VR 2015-04598 and 2019-05384) for generous support.

## Author contributions

A.I. and C.H. conceived and developed the concept and design. J.F., B.G., and A.I. analyzed the data and wrote the manuscript. B.G. supervised experiments involving TReNDs. J.F. performed the experiments. C.H. and A.I. designed and supervised the synthesis of TReNDs. C.P. synthesized the TReNDs and performed LC-MS of FICD and ternary complex. V.P., B.G., and J.F. collected the X-ray diffraction data. V.P. analyzed the X-ray data and refined the structure. C.K. identified AMPylation sites via LC-MS/MS. D.P.-D. performed MD simulations. G.K. supported in protein purification and cell biology. D.H. provided anti-AMP antibodies and guidance on anti-AMP western blotting. M.J.F. supervised the Hsp70 biology and participated in writing the manuscript. M.Z. supervised MD simulations. H.S. supervised LC-MS/MS measurements. A.I. and C.H. provided oversight of the project, data analysis, and data interpretation. All authors participated in manuscript editing and final approval.

## Funding

## Competing interests

The authors declare no competing interests.
