## [Peer Review File · Nature Communications]

REVIEWER COMMENTS

Reviewer #1 (Remarks to the Author):

AMPylation of the ER-resident Hsp70, BiP, by FICD is known to tune its activity, but the structural knowledge of this interaction is not known. Here, the authors use a technology that the authors previously reported to trap the FICD-BiP complex and solve its structure. The important observation is that the TPR domains of FICD are involved and that they are primarily responsible for nucleotide-dependence. This is an interesting observation because it has wider implications for molecular recognition in other FICD and AMPylation functions. More immediately, this structure helps elucidate an interesting and important PTM on BiP. It seems to have the scope and impact that one would expect for Nat. Commun.

However, a few items need to be addressed:

1. It took me some effort to figure out which BiP and FICD construct was being used in each experiment (e.g. BiP 19-654 for biochemistry, 28-549 for structural work). This information can (usually) be gleaned from the legends, but the text might be clearer, especially in the cell-based experiments. The figure panels might also be labelled more carefully to avoid confusion.
2. In relation to this idea, it would be useful to understand the putative role of the “lid” domain. This domain is present in the biochemical experiments but absent from the structure, yet it is known to play important roles in nucleotide cycling. It was difficult for me to discern whether the lid would interfere with (or contribute to) FICD binding?
3. Point mutations are used to test the role of putative contacts in AMPylation. However, the effects of these mutations on BiP structure/function deserve additional attention. BiP is highly conserved and many mutations will damage folding and activity (even if they don't appear to be in structurally important locations). Each of the BiP mutants (e.g. R197A, E217A, B238A, V241A) should be carefully tested by CD, melting temperature and ATPase activity to ensure that the interpretations are correct and not muddled by (currently unseen) alterations in BiP structure. It seems possible that partial unfolding could drive greater/reduced FICD substrate activity by alternative mechanisms.
4. This is a minor comment, but my understanding is that TPR domains usually use either their convex or concave surfaces for protein-protein interactions, but the FICD-BiP contact looks like an unusual “side-on” contact. It might be worth commenting on this and drawing comparisons to similarities and differences.
5. Also a minor comment: where would the transmembrane domain be in relation to the FICD and TPR domains? In other words, what are the constraints on substrate (BiP) entry to the active site that might be imposed by ER membrane?

Reviewer #2 (Remarks to the Author):

The manuscript by Fauser et al reported a beautiful crystal structure of a FICD-BiP complex stabilized by TRENDs, a unique class of thiol-reactive derivatives of ATP, a co-substrate of FICD. Biochemical and computational analysis support structural observations. BiP is the master regulator of protein homeostasis in ER. Recently, it has been demonstrated that the AMPylation of BiP by FICD is an important regulation process on BiP's function. Understanding the molecular mechanism of the AMPylation of BiP by FICD is important for dissecting the regulation of BiP's function in stress response. As shown in a recent publication from the same group in Nature Chemistry, the TRENDs crosslinking is a powerful and novel approach to study substrates for FICD.

However, there are a number of major concerns to prevent it from publishing in its current form.

1. The reported structure is beautiful. However, it seems that the mechanistic insights gained through this structure is limited. This structure revealed that the interaction between FICD and BiP is mainly mediated by the TPR motifs. This is supported by the observation that deleting TRP motifs abolishes the AMPylation of BiP. However, the reported complex structure is forced by crosslinking and was hypothesized to be "a post-catalytic state with the SBD having dissociated from the Fic domain". Very limited interaction between FICD and SBD, the enzymatic site, was observed in this structure, and the authors had to use MD to predict meaningful contacts. Moreover, no direct physical interaction between TPR motif and BiP was assayed in the manuscript. A number of biochemical approaches are suitable for assaying transient interactions such as surface plasmon resonance. Although the manuscript provided mutagenesis support, all the results are AMPylation of BiP or crosslinked complex.

The last section of the results showed that the TPR motifs are important for BiP and EEFLA2, but not UMPS. It's an interesting observation. But, like the rest of the manuscript, the mechanistic implication of this observation is not clear.

2. For the mutational analysis of the TPR-BiP(ATP) contacts, there are a number of the issues with the BiP mutations.

1) R197A has been shown to have an enhanced ATPase activity and compromised NBD-SBD interaction by Henderson lab (Awad et al, PNAS, 2008). Thus, it's hard to interpret the reported AMPylation defect. 2) The authors described that the interaction between TPR and BiP is mainly

through the interdomain linker of BiP (V415 and L417), but did not mutate these residue “since the replacement of residues of the hydrophobic linker would impair the ATP/ADP driven allosteric control of BiP”. This is not true. Mutational analysis at the analogous residues in DnaK (the major E.coli Hsp70) has shown that these residues are not really involved in the allosteric control of Hsp70s (Kumar et al, JMB, 2011). The authors should mutate these residues and test AMPylation and FICD-BiP interaction.

3) The BiP residues that are in the TPR-BiP interfaces are all highly conserved in classic Hsp70s. Does FICD modify cytosolic Hsp70 and Hsc70? Although FICD is mainly in ER, the other two substrates of FICD tested in this manuscript (EEFLA42 and UMPS) are both cytosolic proteins. TPR motifs are also important for EEFLA42. Are there any similarity between BiP and EEFLA42 for TPR recognition? The authors should have a sequence alignment of the interface residues to show conservation.

3. There are a number of issues with the reported ERdj3 tests.

1) First of all, the authors wrote they tested ERdj3 in the Results section. However, after reading the Methods, I realized that only a very small fraction of ERdj3 (28-97) was used. This fraction barely covers the J-domain of ERdj3. The authors should make it clear in the Results that they only used the J-domain of ERdj3.

2) All previous studies have shown that the J-domain alone is not enough to form detectable interaction with Hsp70s although the J-domain is crucial for interacting with Hsp70s.

3) The logic for using BiP-T229A is not well developed. It was reported that the analogous mutant (T199A) in DnaK has no detectable interaction with J proteins.

Thus, it's questionable whether the reported effects in AMPylation are meaningful.

4. For the section on unfolded protein substrates, the authors should confirm whether BiP forms complexes with CH1, the substrate that was tested, in their assays before making any conclusions. Moreover, the authors should test peptide substrates besides CH1.

5. The authors should do a detailed structural comparison of the reported FICD-BiP complex with published Fic-enzymes in complex with their substrates (such as the IbpA-Cdc42 complex). It seems that the TPR-like motifs of IbpA also interacts with Cdc42. Are there any similarities?

minor concerns:

1. For the crystallographic data, the I/σ for the highest resolution shell is 1.1 with very high Rmerge (0.982) and low CC1/2 (0.48), which means the data in this shell is almost meaningless. I would suggest cutting the resolution to a more meaningful limit with reasonable I/σ , Rmerge and CC1/2.

2. For the predicted FICD-BiP-SBD contacts by MD, E514 is on beta7, not on loop78.

Reviewer #3 (Remarks to the Author):

The manuscript entitled “Specificity of AMPylation of the human chaperone BiP is mediated by TPR motifs of FICD” by Fauser provides valuable structural insight into the recognition of substrates by a variant of the dual AMPylating/DeAMPylating enzyme FICD. Using thiol-reactive nucleotide derivatives, the authors were able to solve the crystal structure of a mutant AMPylation competent FICD in complex with its substrate BiP. The author’s structure reveal interaction interfaces between the FICD TPR domains and identified several key interactions for substrate recognition. Overall, this is an important step that provides valuable insight into the substrate recognition by Fic, albeit using a restricted variant FICD. However, a few additional experiments would clarify this observation and increase the significance of their findings.

Remaining questions and concerns:

- 1) Several mutations were present in the FICD construct used for crystallization (FICD 102-445, E234G, E404C, T168A, T183A, L258D). The mutations E234G and L258D have known effects on the AMPylation and deAMPylation activity of FICD. Do mutations in the AMPylation sites T168 and T183 have effects in FICD enzymatic activity?
- 2) The authors crystallized a monomeric version of FICD. However, they comment that the overall structure of FICD’s catalytic domain is not significantly altered with the interaction of BiP’s NBD. A current hypothesis in the field is that FICD becomes monomeric *in vivo* when it’s activity switches from deAMPylator to AMPylator. Does the authors crystal structure indicate any structural corroboration for this hypothesis?
- 3) **Figure 3a,b:** To determine if the interactions between FICD’s TPR domain and BiP were important for substrate recognition, the authors generated several point mutants in the FICD L258D construct and used these double mutants for AMPylation assays. However, FICD L258D itself still retains deAMPylation activity (CITE). Thus, the authors cannot conclude that these mutants have lost their AMPylation activity by simply measuring BiP-AMP by western. It is likely BiP-AMP is being turned over during the assay via deAMPylation. Without establishing that deAMPylation is also being affected, little can be concluded here.
- 4) **Figure 3a and 3f:** The authors do not directly address the same interactions between FICD’s TPR domains and BiP’s NBD are needed for deAMPylation of BiP. The authors should perform deAMPylation assays with the FicD interaction mutants described here. Evidence that FicD does or does not use the same substrate interactions to deAMPylate vs AMPylate would be valuable information in understanding how this enzyme is regulated, and the identification of mutants that further separate these activities is valuable to the

field. The authors touch on this in the discussion (**Lines 455**). With all the tools in hand, it seems like a straightforward next step in their study.

- 5) The authors show that the presence of excess ERdj3 inhibits the AMPylation of BiP by the overactive FICD E234G (**Figure 3c**) and that presence of excess unfolded substrate, CH1 does not alter AMPylation of deAMPylation of BiP (**Figure 4c**). However, both of these systems are highly artificial and tell us little about how FICD would interact with a BiP that is cycling in the ER. A more useful system to study would be to compare AMPylation and deAMPylation of BiP in the presence of stoichiometric ERdj3 with increasing concentrations of CH1.
- 6) **Figure 4e:** The authors found that the TPR domains of FICD are required for AMPylation of EEF1A2. It would be interesting to know if the same residues in the TPR domain required for recognition of BiP (K121, K124) are also required for EEF1A2.
- 7) **Line 448:** The authors make a point to mention that the monomeric FICD has a nearly identical structure that that of previously published dimeric FICD “indicating that the proposed conformational plasticity of the Fic domain is difficult to grasp via crystallography”. In this discussion, the authors do not take into consideration the other mutations they have added to their crystallized construct, namely E234G that dispels a critical salt bridge in the FICD domain active site, resulting in artificial activation of the AMPylation activity and abolishment of the deAMPylation activity of this enzyme . Since others have indicated that the monomeric form of FICD allows this salt bridge to become more malleable, it is not surprising that little changes are observed in the structure where this interaction is not present.
- 8) Furthermore, the authors should acknowledge that FICD is a membrane associated protein and thus its movement and dimerization may show different qualities *in vivo*.
- 9) **Line 484:** The authors theorize that FICD localization could change from within the ER to cytoplasmic “upon certain stressors or in different cell lines and tissues.” Has such a change in localization of a membrane protein been described before? A reference to such an example would be beneficial in understanding how a such a topological change is possible.
- 10) In addition, recent work has shown that EEF1A2 and UMPS AMPylation were not dependent on the expression of FICD in the cell as loss of FICD did not change observed AMPylation levels of these proteins. This is in contrast to BiP AMPylation which is FICD dependent. (DOI: 10.1038/s41467-019-14235-6) A more likely possibility is another cytosolic enzyme is responsible for AMPylation of EEF1A2 and UMPS which should be addressed in the discussion.

- 11) Line 440:** Multiple AMPylation sites on BiP have been reported by several research groups. The authors comment that when BiP T518A was used as a substrate, no FICD^{TREND2}-BiP ternary complex formation occurred. They argue that this suggests T518 is the only valid BiP AMPylation site. However, BiP T518A has previously been shown to be not fully functional as it not viable in flies. Importantly the BiP T366A mutation has been found to phenocopy *fic* knockouts in flies (DOI: [10.7554/eLife.38752](https://doi.org/10.7554/eLife.38752)). Furthermore, the authors state that they are also using the BiP 229A mutant which biases BiP to an ATP bound conformation and results in a steric block for access to the T366 site. It is possible that a BiP T518A mutant biases BiP to a conformation that blocks FICD recognition of other AMPylation sites. If AMPylated BiP is used as a substrate, can additional FICD^{TREND2}-BiP ternary complex formations occur? Do FICD^{TREND2}-BiP ternary complex formations occur? If wildtype BiP is used as a substrate, are there multiple populations of FICD^{TREND2}-BiP ternary complex formations?
- 12) Line 350:** Fig. 4b, Bip^{AMP} in the presence of ADP showed an increase and then decrease of AMPylation. Is this accidental in the figure or a consistent result? This should be discussed.
- 13) Fig 2a:** Though noted in the legend, the figure can be labeled for better illustration, including the domains and nucleotides. The linker can be colored differently as currently it is not obvious.
- 14) Line 183:** “Cα position of M135” can be shown in the Fig. 2c.
- 15) Line 196-197:** FICD also interacts with SBD as shown in Fig. 2b. This should be mentioned.
- 16) Fig. S4b:** The time point “60” should be “1” if “t in h” is the unit.
- 17) Line 276:** What is “QPD”? This is not explained and confusing.
- 18)** Supplemental figures should be cited more accurately including “a, b, c...”. It’s hard to find the exact figure panels the authors refer to as there are many in each figure and they are not cited in sequential numbers.
- 19)** The rotation signs (Fig. 2c and Fig. S3g) are confusing, as they can be interpreted from both directions. Instead a complete oval with an arrowhead should be used with partial of the straight line hidid.

Reviewer #1

AMPylation of the ER-resident Hsp70, BiP, by FICD is known to tune its activity, but the structural knowledge of this interaction is not known. Here, the authors use a technology that the authors previously reported to trap the FICD-BiP complex and solve its structure. The important observation is that the TPR domains of FICD are involved and that they are primarily responsible for nucleotide-dependence. This is an interesting observation because it has wider implications for molecular recognition in other FICD and AMPylation functions. More immediately, this structure helps elucidate an interesting and important PTM on BiP. It seems to have the scope and impact that one would expect for Nat. Commun.

We appreciate the reviewer's critical evaluation of our manuscript and thank for the positive reception of the manuscript.

However, a few items need to be addressed:

1. It took me some effort to figure out which BiP and FICD construct was being used in each experiment (e.g. BiP 19-654 for biochemistry, 28-549 for structural work). This information can (usually) be gleaned from the legends, but the text might be clearer, especially in the cell-based experiments. The figure panels might also be labelled more carefully to avoid confusion.

We thank the reviewer for this suggestion. We have improved the text and the labeling accordingly. We now define the used constructs more clearly in the beginning of the manuscript:

Line 123: If not stated differently, the FICD construct aa 102-458 was used for all experiments as this version yielded a homogenous protein. Various cysteine substitutions (H319C_{FICD}, L403C_{FICD}, E404C_{FICD}, N407C_{FICD}) within the ATP binding pocket of the constitutively active FICD 102-458 E234G were tested for their chemical coupling with TReNDs (**Fig. 1c**).

Line 137: We then tested whether FICD 102-445 E234G E404C^{TReND-2} forms a covalently linked complex upon BiP addition in the presence of Mg²⁺. If not stated differently, BiP was employed as near full-length version aa 19-654 for all experiments, lacking only the N-terminal signal peptide.

Line 148: We then sought to form the covalently linked complex at a preparative scale for structural analysis. For this purpose, we chose FICD 102-445 and BiP 28-549 as crystallization constructs, since they have been previously crystallized successfully.^{20,23}

Figure 1g:

Line 166: **Fig. 1. Binary adduct formation and complex formation. ... d)** Reactivity of TReNDs towards FICD 102-458 E234G bearing cysteine substitutions resolved by Phos-TagTM SDS-PAGE. TReND-mix consists of TReND-1, TReND-2, and TReND-3 at equimolar concentration ... **g)** SDS-PAGE of the purified ternary complex (using BiP 28-549 T229A and FICD 102-445 T168A T183A E234G L258D E404C) submitted for crystallography.

In the text we specifically mention the constructs used for the cell-based validation (see below Line 281). For clarity, we adjusted the labelling in the figure legend (see below Line 287).

Line 309: To verify the contribution of the TPR motifs to BiP AMPylation within living cells, we overexpressed FICD 102-458 E234G and FICD 187-458 E234G (Δ TPR) in HEK293 cells and monitored BiP-AMPylation via western blotting (**Fig. 3d**).

Line 317: **Fig. 3. Confirmation of the TPR-NBD interface of the FICD^{TReND}:BiP complex and biochemical elucidation of Fic domain-SBD interaction. ... d)** Validation of the contribution of the TPR motif to BiP AMPylation in HEK293 cells. The lysates of HEK293 cells overexpressing different constructs of GFP-tagged FICD with (102-458) or without TPR motifs (Δ TPR: 187-458) were analyzed via western blotting. GFP was detected to verify expression, whereas GAPDH detection served as a loading control. Representative blots are shown from three biological replicates.

Line 372: It was previously reported that FICD AMPylates the ATP-bound state of BiP.¹⁹ We confirmed and extended this finding by monitoring the kinetics of ternary complex formation between BiP and FICD^{TReND-2} (102-445 E234G E404C) in the presence of ADP, ATP or no nucleotide (**Fig. 4a and Supplementary Fig. 11**).

Line 392: **Fig. 4. Effect of nucleotides, unfolded proteins, and TPR motifs on FICD mediated AMPylation and deAMPylation...** **c)** AMPylation of BiP with (+ C_{H1}) and without (- C_{H1}) the unfolded protein substrate C_{H1}. AMPylation of BiP in the ATP-bound state of BiP was monitored in an AMPylation assay and analyzed via western blot using the α AMP antibody (upper panel). AMPylation of BiP 19-654 WT in its ADP bound state was monitored via ternary complex formation using FICD 102-445 E234G E404C (lower panel). The full SDS-PAGE gels and the silver stained western blots are deposited in **Supplementary Fig. 13**.

Line 427: To assess a possible effect of C_{H1} on the AMPylation of BiP:ADP, we monitored the influence of C_{H1} on covalent ternary complex formation of FICD 102-445 E234G E404C and BiP, in which the use of TReND permits the analysis of “AMPylation” of BiP:ADP.

2. In relation to this idea, it would be useful to understand the putative role of the “lid” domain. This domain is present in the biochemical experiments but absent from the structure, yet it is known to play important roles in nucleotide cycling. It was difficult for me to discern whether the lid would interfere with (or contribute to) FICD binding?

For crystallization of the complex we employed a lidless version of BiP as this construct proved suitable for structural characterization of Hsp70s in previous studies (Chang et al., 2008; Preissler et al., 2017).

Both BiP and BiP Δ lid can be AMPylated by FICD. Our biochemical and structural work shows that FICD binds to BiP in a BiP:ATP-like conformation by interacting with both NBD and SBD. The conformation of BiP:ATP is well defined (as judged by crystallography (Yang et al., 2015) and in-solution techniques such as single molecule FRET (Marcinowski et al., 2011)) with the lid detached from the SBD and tucked onto the NBD opposite to the FICD–BiP interfaces. Superimposition of BiP:ATP with complexed BiP (see below) indicates that the lid does not directly interfere with FICD binding.

Figure legend: Superimposition of BiP:ATP (PDB 5E84) with complexed BiP. Complexed BiP is displayed in orange, FICD in purple, and BiP:ATP in grey.

3. Point mutations are used to test the role of putative contacts in AMPylation. However, the effects of these mutations on BiP structure/function deserve additional attention. BiP is highly conserved and many mutations will damage folding and activity (even if they don't appear to be in structurally important locations). Each of the BiP mutants (e.g. R197A, E217A, B238A, V241A) should be carefully tested by CD, melting temperature and ATPase activity to ensure that the interpretations are correct and not muddled by (currently unseen) alterations in BiP structure. It seems possible that partial unfolding could drive greater/reduced FICD substrate activity by alternative mechanisms.

We agree with the reviewer that the additional experiments on the integrity of BiP mutants are critical for thorough evaluation of FICD substrate activity towards BiP mutants. We therefore determined the far-UV CD spectra, the melting temperatures, and the steady state ATPase kinetics of all BiP-mutants, summarized in Supplementary Fig. 7. Overall, the mutants displayed not notable differences with the following exceptions: BiP E217A shows an increased melting temperature of the NBD and decreased ATPase activity and BiP R197A has accelerated ATP hydrolysis (as previously reported (Awad et al., 2008)). Regarding the residues involved in the interaction with FICD's Fic domain we observed that the melting temperatures of the SBD of BiP E514A and BiP K516A were clearly reduced.

We highlighted these findings in the results section and briefly discussed the potential impact on AMPylation efficiency.

We have changed the result section as follows.

Line 268: In order to exclude that the observed changes in AMPylation result from structural alterations, we determined the melting temperatures by NanoDSF, the steady state ATPase kinetics using an enzyme coupled assay, and potential structural changes by circular dichroism (CD) spectroscopy for each BiP-mutant (Supplementary Fig. 7). All BiP mutants exhibited properties similar to BiP WT with two exceptions: BiP E217A shows an increased melting temperature of the NBD⁴¹ together with reduced ATPase activity; BiP R197A has accelerated ATP hydrolysis as reported previously.⁴² Since both E105A_{FICD} and R197A_{BiP} resulted in reduced BiP AMPylation to a similar extent (~80%) it is likely that the loss of the ionic bond of E105_{FICD} - R197_{BiP} is mainly responsible for the observed AMPylation change rather than the reported effects of R197A_{BiP} on ATP hydrolysis and NBD-SBD interaction.⁴²

41. Lamb, H. K. *et al.* The Affinity of a Major Ca²⁺ Binding Site on GRP78 Is Differentially Enhanced by ADP and ATP. *J Biol Chem* **281**, 8796–8805 (2006).

42. Awad, W., Estrada, I., Shen, Y. & Hendershot, L. M. BiP mutants that are unable to interact with endoplasmic reticulum DnaJ proteins provide insights into interdomain interactions in BiP. *Proc. Natl. Acad. Sci. U. S. A.* **105**, 1164–1169 (2008).

Line 347: Of note, the melting temperatures of the SBD were clearly reduced upon substitution of E514_{BiP} and K516_{BiP} by alanine (Supplementary Fig. 7). Hence, their observed contribution to BiP AMPylation may be partially due to structural perturbation of the SBD.

Supplementary Fig. 7

Supplementary Fig. 7. Biochemical and biophysical characterization of the BiP mutants. **a)** The melting points of the BiP constructs was determined via NanoDSF. Two distinct melting points (T_{m1} for NBD and T_{m2} for SBD) were observed as displayed in the first derivative of the fluorescence ratio 350/330 nm (left panel). The melting points of all BiP constructs are summarized in the middle (T_{m1}) and right panel (T_{m2}). **b)** Determination of steady state ATPase kinetics were obtained using an ATP regenerating system. ATP hydrolysis results in the consumption of NADH yielding a negative k_{cat} value. **c)** For all BiP constructs a CD spectrum ranging from 185 to 260 nm was obtained. The ellipticity θ is displayed in the unit of mdeg. All experiments were performed as triplicates. The error bars represent the standard mean deviation.

Methods:

Line 723: Melting point determination via nano dynamic scanning fluorimetry (NanoDSF)

All BiP mutants were diluted to 1 mg/mL in HKM buffer. The samples were loaded into the standard capillaries (nanotemper, #PR-C002) and analyzed via the Prometheus NT.48 (nanotemper) at a gradient of 1 °C/min ranging from 20-80 °C. The melting points were derived from the ratio of fluorescence at 350/330 nm.

Steady state ATPase assay

The ATPase rate of BiP and BiP mutants was determined applying an ATP regenerating system in which the consumption of NADH directly corresponds to the hydrolysis of ATP.⁷⁶ The absorption of NADH was measured at 340 nm in a TECAN Spark microplate reader (Tecan) in Corning® 384 well plates (#3640). The assay was performed at 37 °C, 1 mM ATP and 2 µM BiP.

Circular dichroism spectroscopy

All BiP mutants were diluted to 0.1 mg/mL in 10 mM potassium phosphate pH 7.5, 150 mM KF, 1 mM MgCl₂. The spectra were collected at 25 °C in a 0.1 mm cuvette from 185 – 260 nm on a Chirascan CD spectrometer (Applied Photophysics). The instrument was set to 0.5 nm bandwidth, 0.5 s response and 0.5 nm data pitch. The spectra were analyzed with Pro-Data Viewer and background subtracted. Each curve represents the mean of three measurements.

76. Nørby, J. G. Coupled Assay of Na⁺,K⁺-ATPase Activity. Methods Enzymol. 156, 116–119 (1988).

4. This is a minor comment, but my understanding is that TPR domains usually use either their convex or concave surfaces for protein-protein interactions, but the FICD-BiP contact looks like an unusual “side-on” contact. It might be worth commenting on this and drawing comparisons to similarities and differences.

As suggested, we have extended our discussion on this matter as follows:

Line 516: Of note, this interaction mode is fundamentally different from other Hsp70 associated proteins that bear TPR-motifs, in which the concave surface of the TPR-domain specifically binds to the C-terminal peptide of Hsp70.^{54,55} Interestingly, the herein described “side-on” TPR interaction interface has been described before⁵⁶, yet presents a rather rare interaction mode of TPR motifs that usually interact via their convex or concave surfaces.⁵⁷

56. Quinaud, M. et al. Structure of the heterotrimeric complex that regulates type III secretion needle formation. PNAS 104, 7803–7808 (2007).

57. Zeytuni, N. & Zarivach, R. Structural and Functional Discussion of the Tetra-Trico-Peptide Repeat, a Protein Interaction Module. Structure 20, 397–405 (2012).

Screenshot (not included in the manuscript): Superimposition of the TPR motifs of FICD:BiP complex structure (BiP white, FICD green) with TPR motifs of PscG (blue), another TPR-motif containing protein together with its interaction partners PscE (red) and PscF (grey) (PDB: 2UWJ). While PscF binds to the concave surface of the TPR motif, PscE binds (similar to FICD:BiP complex) to the side of the TPR motif.

5. Also a minor comment: where would the transmembrane domain be in relation to the FICD and TPR domains? In other words, what are the constraints on substrate (BiP) entry to the active site that might be imposed by ER membrane?

In order to address this comment, we added the following statement in the discussion section. For illustration purposes, we additionally included a figure in the supplementary materials.

Discussion:

Line 483: Importantly, FICD is anchored to the ER membrane via its transmembrane domain.^{15,52} A stretch of more than 50 partially unstructured amino acids (JPred⁵³) between the transmembrane domain and the BiP-engaging TPR motifs would allow the herein proposed mode of interaction of FICD and BiP (Supplementary Fig. 14).

52. Rahman, M. *et al.* Visual neurotransmission in *Drosophila* requires expression of Fic in glial capitate projections. *Nat. Neurosci.* **6**, 871–875 (2012).

53. Drozdetskiy, A., Cole, C., Procter, J. & Barton, G. J. JPred4: A protein secondary structure prediction server. *Nucleic Acids Res.* **43**, W389–W394 (2015).

Supplementary Information:

Supplementary Fig. 14. Schematic representation of the FICD:BiP complex associated to the ER membrane. BiP (orange) is bound to FICD (blue). The N-terminal part of FICD is predicted to bear a cytoplasmic tail, followed by a transmembrane domain that anchors the protein to the ER membrane (Uniprot). In between the transmembrane domain and the TPR motif that is involved in BiP recognition are more than 50 amino acids (aa), partially structured as helix, partially unstructured as predicted by JPred⁵³.

Reviewer #2

The manuscript by Fauser et al reported a beautiful crystal structure of a FICD-BiP complex stabilized by TReNDs, a unique class of thiol-reactive derivatives of ATP, a co-substrate of FICD. Biochemical and computational analysis support structural observations. BiP is the master regulator of protein homeostasis in ER. Recently, it has been demonstrated that the AMPylation of BiP by FICD is an important regulation process on BiP's function. Understanding the molecular mechanism of the AMPylation of BiP by FICD is important for dissecting the regulation of BiP's function in stress response. As shown in a recent publication from the same group in Nature Chemistry, the TReNDs crosslinking is a powerful and novel approach to study substrates for FICD.

We appreciate the reviewer's critical evaluation of our manuscript and thank for the kind words.

However, there are a number of major concerns to prevent it from publishing in its current form.

1. The reported structure is beautiful. However, it seems that the mechanistic insights gained through this structure is limited. This structure revealed that the interaction between FICD and BiP is mainly mediated by the TPR motifs. This is supported by the observation that deleting TRP motifs abolishes the AMPylation of BiP. However, the reported complex structure is forced by crosslinking and was hypothesized to be "a post-catalytic state with the SBD having dissociated from the Fic domain". Very limited interaction between FICD and SBD, the enzymatic site, was observed in this structure, and the authors had to use MD to predict meaningful contacts. Moreover, no direct physical interaction between TPR motif and BiP was assayed in the manuscript. A number of biochemical approaches are suitable for assaying transient interactions such as surface plasmon resonance. Although the manuscript provided mutagenesis support, all the results are AMPylation of BiP or crosslinked complex.

The last section of the results showed that the TPR motifs are important for BiP and EEFLA2, but not UMPS. It's an interesting observation. But, like the rest of the manuscript, the mechanistic implication of this observation is not clear.

We thank the reviewer for the comment. To date the structural investigation of the AMPylation complex of FICD and BiP remained elusive since it proved highly challenging to produce and isolate this complex. In order to overcome this limitation, we set out to apply nucleotide analogues for covalently tethering FICD to BiP. We agree with the reviewer that this is an artificial linkage yet it was necessary for the aforementioned reasons. Moreover, we showed previously, that the nucleotide analogues (TReNDs) are bound by Fic enzymes very similar to endogenous ATP and do not alter the structure of enzyme-substrate complexes. Importantly, the herein applied crosslinking concept relies on enzymatic activity which makes a productive (and therefore native-like) binding event mandatory for the covalent crosslink formation. As pointed out by the reviewer, even the covalent linkage of FICD and BiP did not prevent partial dissociation of the Fic-domain and SBD which reflects the challenges in obtaining and isolating this AMPylation complex. In line with our structural analysis, we observed that FICD is unable to AMPylate different constructs of the isolated SBD which underlines the importance of the TPR-NBD interface for the complex formation of FICD and BiP.

The manuscript provides mechanistic insights into AMPylation of BiP by demonstrating how binding of the TPR motifs mediate specificity towards the domain-docked conformation of BiP. Furthermore, we were able to show that a minor twist of the TPR motifs and not a major reorientation contributes to BiP binding and AMPylation (as elaborated in comment 2 of reviewer 3). Inspired by reviewer 3 (comment 4), we additionally provided evidence to what extent the interactions that are involved in BiP AMPylation also contribute to BiP deAMPylation.

We value the reviewer's suggestion of additional control experiments on the validation of the TPR-NBD interface by assaying the physical interaction of FICD and BiP and therefore performed additional experiments. So far, we extensively validated the TPR interface and the involved interaction partners by cell biological experiments as well as *in vitro* AMPylation assays. Although we think that the drawn conclusions are well-founded on the so far performed experiments, we decided to assay the direct interaction of FICD and BiP by several methods such as isothermal titration calorimetry (ITC), bio-layer interferometry (BLI), and microscale thermophoresis (MST). However, all these attempts failed due to the limitations in protein stability of FICD at high molecular concentrations and compatible buffer conditions (ITC) or persistent unspecific binding as determined by appropriate controls in BLI and MST. Eventually, we added experimental data of size-exclusion chromatography to the manuscript that suggests that binding of FICD to BiP occurs and is dependent on the TPR motifs as demonstrated by the other assays.

Line 277: **Noteworthy, FICD E234G lacking the TPR motifs (Δ TPR) was devoid of AMPylation activity, highlighting the general importance of the TPR motifs for BiP binding (Supplementary Fig. 5). An observation that is supported by data obtained by size-exclusion chromatography which suggests that complex formation is dependent on the TPR motifs (Supplementary Fig. 8).**

Line 720: The binding of FICD to BiP was analyzed by incubating 5 μ M FICD L258D 102-458 with 100 μ M BiP 19-654 T229A T518A in HKM buffer supplemented with 5 mM ATP for at least two hours. The proteins were separated on a Superdex 10/300 75 pg using the same parameters and buffer as described above.

Supplementary Fig. 8

Supplementary Fig. 8. The TPR motifs are important for BiP binding. 5 μ M FICD L258D with (102-458, left panel) and without (187-458, right panel) the TPR motifs were added to

100 μ M BiP 19-654 T229A T518A in presence of 5 mM ATP. The spectra suggest that FICD binding to BiP is dependent on the TPR motifs.

2. For the mutational analysis of the TPR-BiP(ATP) contacts, there are a number of the issues with the BiP mutations.

1) R197A has been shown to have an enhanced ATPase activity and compromised NBD-SBD interaction by Henderson lab (Awad et al, PNAS, 2008). Thus, it's hard to interpret the reported AMPylation defect.

As discussed in the manuscript, the complex structure revealed a salt bridge of R197_{BiP} with E105_{FICD} which may be responsible for the observed AMPylation defect that we observed upon alanine substitution of both R197A_{BiP} and E105A_{FICD}. Interestingly, the alanine substitutions of both proteins reduce the AMPylation of BiP to a similar extent (~80%) which suggests that the loss of interaction (and not structural alterations of BiP upon mutation of R197) is mainly responsible for the observed AMPylation defect. However, we do understand the reviewer's concern and therefore added a corresponding statement in the results section and furthermore added data on steady state ATPase kinetics of all BiP mutants. As mentioned above (Reviewer 1, Comment 3) we additionally characterized all BiP mutants by CD spectroscopy and melting temperature.

Supplementary Fig. 7. Biochemical and biophysical characterization of the BiP mutants.

a) The melting points of the BiP constructs was determined via NanoDSF. Two distinct melting points (T_{m1} and T_{m2}) were observed as displayed in the first derivative of the fluorescence ratio 350/330 nm (left panel). The melting points of all BiP constructs are summarized in the middle (T_{m1}) and right panel (T_{m2}). **b)** Determination of steady state ATPase kinetics were obtained using an ATP regenerating system. ATP hydrolysis results in the consumption of NADH yielding a negative k_{cat} value. **c)** For all BiP constructs a CD spectrum ranging from 185 to 260 nm was obtained. The ellipticity θ is displayed in the unit of mdeg. All experiments were performed as triplicates. The error bars represent the standard mean deviation.

Results:

Line 268: In order to exclude that the observed changes in AMPylation result from structural alterations, we determined the melting temperatures by NanoDSF, the steady state ATPase kinetics using an enzyme coupled assay, and potential structural changes by circular dichroism (CD) spectroscopy for each BiP-mutant (Supplementary Fig. 7). All BiP mutants exhibited properties similar to BiP WT with two exceptions: BiP E217A shows an increased melting temperature of the NBD⁴¹ together with reduced ATPase activity; BiP R197A has accelerated ATP hydrolysis as reported previously.⁴² Since both E105A_{FICD} and R197A_{BiP} resulted in reduced BiP AMPylation to a similar extent (~80%) it is likely that the loss of the

ionic bond of E105_{FICD} - R197_{BiP} is mainly responsible for the observed AMPylation change rather than the reported effects of R197A_{BiP} on ATP hydrolysis and NBD-SBD interaction.⁴²

41. Lamb, H. K. *et al.* The Affinity of a Major Ca²⁺ Binding Site on GRP78 Is Differentially Enhanced by ADP and ATP. *J Biol Chem* **281**, 8796–8805 (2006).

42. Awad, W., Estrada, I., Shen, Y. & Hendershot, L. M. BiP mutants that are unable to interact with endoplasmic reticulum DnaJ proteins provide insights into interdomain interactions in BiP. *Proc. Natl. Acad. Sci. U. S. A.* **105**, 1164–1169 (2008).

Methods:

Line 728: **Steady state ATPase assay**

The ATPase rate of BiP and BiP mutants was determined applying an ATP regenerating system in which the consumption of NADH directly corresponds to the hydrolysis of ATP.⁷⁶ The absorption of NADH was measured at 340 nm in a TECAN Spark microplate reader (Tecan) in Corning® 384 well plates (#3640). The assay was performed at 37 °C, 1 mM ATP and 2 μM BiP.

76. Nørby, J. G. Coupled Assay of Na⁺,K⁺-ATPase Activity. *Methods Enzymol.* **156**, 116–119 (1988).

2) The authors described that the interaction between TPR and BiP is mainly through the interdomain linker of BiP (V415 and L417), but did not mutate these residue “since the replacement of residues of the hydrophobic linker would impair the ATP/ADP driven allosteric control of BiP”. This is not true. Mutational analysis at the analogous residues in DnaK (the major E.coli Hsp70) has shown that these residues are not really involved in the allosteric control of Hsp70s (Kumar et al, JMB, 2011). The authors should mutate these residues and test AMPylation and FICD-BiP interaction.

We apologize for this oversight and thank the reviewer for pointing this out. As requested, we now analyzed the V415A and L417A mutants and analyzed their substrate properties by AMPylation assays and included these constructs in all control experiments (CD spectroscopy, ATPase assay, melting temperature). The loss of AMPylation upon alanine substitution of V415_{BiP} (~by 70%) and L417_{BiP} (no AMPylation) further supported the suggested mode of interaction and corroborated the relevance of the hydrophobic cluster. Importantly, both BiP alanine substitutions proved indifferent to BiP WT in CD, ATPase assay and melting temperature. Figure 3b:

Line 261: The significance of the hydrophobic cluster was probed by alanine substitution of V241_{BiP}, and the two linker residues V415_{BiP} and L417_{BiP} that are not involved in the allosteric control of Hsp70s.⁴⁰ Indeed, we found that the substitution of V241_{BiP} decreased AMPylation by more than 95% and substitution of V415 by 70%. No AMPylation of BiP L417A was observed further corroborating the relevance of the hydrophobic cluster for FICD-BiP interaction.

40. Kumar, D. P. *et al.* The four hydrophobic residues on the Hsp70 inter-domain linker have two distinct roles. *J. Mol. Biol.* **411**, 1099–1113 (2011).

3) The BiP residues that are in the TPR-BiP interfaces are all highly conserved in classic Hsp70s. Does FICD modify cytosolic Hsp70 and Hsc70? Although FICD is mainly in ER, the other two substrates of FICD tested in this manuscript (EEFLA42 and UMPS) are both cytosolic proteins. TPR motifs are also important for EEFLA42. Are there any similarity between BiP and EEFLA42 for TPR recognition? The authors should have a sequence alignment of the interface residues to show conservation.

Our observation is that the TPR motif of FICD recognizes a patch within BiP spanning multiple secondary structure elements rather than a specific sequence (see below, taken from Fig 2f).

Therefore, producing a specific TPR-binding sequence from BiP is not possible. However, we extracted the BiP-sequences TIDNGVFE and QAGVLSGDQDTGDLVLL that provide the longest coherent binding patches and submitted them to BLAST analysis with EEFLA2.

However, no significant homologies have been found and we thus anticipate that the mode of EEF1A binding to FICD must be different from BiP.

However, inspired by the reviewers' comment, we tested and confirmed AMPylation of Hsp70 and Hsc70 by FICD. We added this data together with sequence alignments of BiP and FICD of different model organisms and illustrate that the interacting residues are conserved and thus the mode of interaction is likely to be conserved, too.

Supplementary Information:

Supplementary Fig. 9

Supplementary Fig. 9 The binding mode of the TPR motifs of FICD is conserved. a) Sequence alignment of FICD and BiP from different species using the Clustal Omega Multiple Sequence Alignment tool. The interacting residues within FICD and BiP are highlighted in blue (hydrophilic interactions) and grey (hydrophobic interactions). **b)** FICD AMPylates human BiP, Hsp70 and Hsc70 as detected by western blot with an AMP-specific antibody. The reaction was stopped after 2 h by addition of Laemmli buffer.

Results:

Line 282: In addition, we performed a sequence alignment of FICD and BiP with their homologues from different model organisms. The alignment illustrates that the interacting residues are conserved and thus the mode of interaction is likely to be conserved, too (**Supplementary Fig. 9**). This view is supported by AMPylation of the related chaperones human Hsp70 and Hsc70 by FICD in vitro (**Supplementary Fig. 9**).

Discussion:

Line 494: Moreover, the residues involved in the interface of the TPR motif and NBD are conserved among different species, suggesting that the mode of interaction and the targeted residue T518_{BiP} is conserved, too.

Acknowledgements:

Line 1029: We thank Johannes Buchner at TU Munich for sharing the plasmids of BiP, C_{H1} and ERdj3 and providing Hsp70 and Hsc70.

3. There are a number of issues with the reported ERdj3 tests.

1) First of all, the authors wrote they tested ERdj3 in the Results section. However, after reading the Methods, I realized that only a very small fraction of ERdj3 (28-97) was used. This fraction barely covers the J-domain of ERdj3. The authors should make it clear in the Results that they only used the J-domain of ERdj3.

We understand the reviewers concern and therefore highlighted the used ERdj3 construct in the results section. The AMPylation assays in presence of the J-domain served to further validate the complex structure since the J-domain itself binds to the NBD in a similar way as FICD (Kityk et al., 2018). Therefore, we chose to work with the isolated J-domain to assess a potential competition of FICD and the J-domain for this specific NBD interface.

Line 291: *Since this region is also recognized by the J-domain of Hsp40 cochaperones such as ERdj3, we wondered whether the presence of the J-domain (ERdj3 28-97, referred to as ERdj3 hereafter) would interfere with the AMPylation of BiP⁴³.*

Line 317: **Fig. 3. Confirmation of the TPR-NBD interface of the FICD^{TReND}:BiP complex and biochemical elucidation of Fic domain-SBD interaction. ... c) AMPylation of 10 μ M BiP by FICD 102-458 E234G (0.1 μ M) in the presence or absence of the J-domain of ERdj3 28-97 (200 μ M).**

2) All previous studies have shown that the J-domain alone is not enough to form detectable interaction with Hsp70s although the J-domain is crucial for interacting with Hsp70s.

We agree with the reviewer that the J-domain alone is not enough to stably interact with Hsp70s. Therefore, we used an excess of the J-domain to effectively compete with FICD for BiP binding. It has been shown that the isolated J-domain is sufficient for stimulation of ATP hydrolysis of BiP (Marcinowski et al., 2011) and BiP oligomerization (Preissler et al., 2017) in vitro indicating that binding occurs, albeit transiently. Since the J-domain itself binds to the same surface patch on the NBD as FICD (Kityk et al., 2018), we decided to investigate the suggested binding mode of the TPR to the NBD by analyzing the competition of FICD and the J-domain for BiP. Indeed, we observed a competition of FICD and the J-domain in the AMPylation assay, supporting our mutagenesis study.

3) The logic for using BiP-T229A is not well developed. It was reported that the analogous mutant (T199A) in DnaK has no detectable interaction with J proteins. Thus, it's questionable whether the reported effects in AMPylation are meaningful.

According to this comment, we discuss the concerns in the results section (see below). We agree that this experiment alone cannot verify the binding mode of FICD to BiP. We rather performed this experiment in addition to our other experiments (alanine substitutions, cell biology) for validation of our crystal structure. And importantly, it does support the observed and further validated binding mode of the TPR motif.

We agree that the interaction of DnaK T199A with DnaJ was not detectable in previous studies (Mayer et al., 1999; Suh et al., 1999). However, DnaK T199A was also used to successfully characterize the structure of J-domain bound to DnaK T199A (Kityk et al., 2018) suggesting that DnaK T199A is in principal binding to the J-domain, albeit with low affinity. We sought to make use of the T229A mutant to minimize ATP hydrolysis in order to verify

that reduction in AMPylation is due to FICD/J-domain competition and not due to the conversion of BiP:ATP to BiP:ADP (which is a worse FICD AMPylation substrate).

Line 300: Even though the bacterial BiP T229A homologue, DnaK T199A, has no detectable binding with J-proteins in vitro^{45,46}, we observed that FICD mediated AMPylation of BiP T229A was reduced in the presence of large excess of ERdj3 HPD by approximately 60%, consistent with the transient interaction of ERdj3 with BiP in its ATP-bound state (Fig. 3c)^{10,47}. This effect was more dominant regarding AMPylation of BiP WT due to J-protein stimulated ATP-hydrolysis. Interestingly, ERdj3 QPD also inhibited BiP-AMPylation, albeit to a lower extent, suggesting residual binding of ERdj3 QPD to BiP. Thus, in addition to the alanine substitution experiments, the competition of J-protein with FICD mediated BiP AMPylation further supports the validity of the observed complex crystal structure.

45. Mayer, M. P., Laufen, T., Paal, K., McCarty, J. S. & Bukau, B. Investigation of the interaction between DnaK and DnaJ by surface plasmon resonance spectroscopy. *J. Mol. Biol.* **289**, 1131–1144 (1999).
46. Suh, W. C., Lu, C. Z. & Gross, C. A. Structural features required for the interaction of the Hsp70 molecular chaperone DnaK with its cochaperone DnaJ. *J. Biol. Chem.* **274**, 30534–30539 (1999).

4. For the section on unfolded protein substrates, the authors should confirm whether BiP forms complexes with CH1, the substrate that was tested, in their assays before making any conclusions. Moreover, the authors should test peptide substrates besides CH1.

In order to confirm C_{H1} binding to BiP, we labeled C_{H1} with the fluorescent dye FITC and incubated it with BiP:ADP. Using size-exclusion chromatography, we were able to determine binding of FITC-labeled C_{H1} to BiP as reported before (Feige et al., 2009; Marcinowski et al., 2011). Importantly, we observed that the ability of AMPylated BiP (BiP^{AMP}) to bind unfolded protein substrates is impaired compared to unmodified BiP which explains the indifference of BiP^{AMP} deAMPylation in presence and absence of C_{H1} (**Supplementary Fig. 12**). This observation is in agreement with the literature where modified BiP was found to be underrepresented in complex to protein substrates (Hendershot et al., 1988). We included the control experiments regarding C_{H1} binding as asked in the supplements and extended the discussion accordingly.

The binding of BiP to peptides stimulates ATP hydrolysis (Mayer et al., 2003) which makes it difficult to determine whether the observed change in AMPylation is due to accumulation of BiP:ADP or other minor structural rearrangements. In addition, peptide substrates do not represent physiological targets of BiP, protein substrates also binding to the lid and inducing a different conformation than peptide substrates (Marcinowski et al., 2011), which is why we decided not to include this experiment.

Supplementary Fig. 12

Supplementary Fig. 12. Binding of BiP to the unfolded protein substrate C_{H1} is impaired upon AMPylation. **a)** Left panel: Detailed view on the SBD of complexed BiP (orange) overlaid to BiP:ATP (green, PDB: 5E84). The cleft where protein substrates bind is open and indicated by the red star. Right panel: Position of the substrate binding cleft in the complex structure. The cleft where protein substrates bind is indicated by the red star. **b)** Binding of unmodified BiP (10 μ M) to the unfolded protein substrate C_{H1} (2.5 μ M). The upper panel displays BiP trace at 280 nm whereas the lower panel indicates BiP binding to FITC labeled- C_{H1} at 496 nm. **c)** Binding of AMPylated BiP (BiP^{AMP}) (10 μ M) to the unfolded protein substrate C_{H1} (2.5 μ M). The upper panel displays BiP^{AMP} trace at 280 nm whereas the lower panel indicates BiP^{AMP} binding to FITC-labeled C_{H1} at 496 nm.

Based on the findings we extended the results, methods and discussion section as follows.

Results:

Line 390: However, addition of C_{H1} did not accelerate ternary complex formation, suggesting BiP:ADP bound to C_{H1} is not a favored AMPylation substrate despite obtaining the domain-docked conformation¹⁰. The reported K_D of 7.4 μM, suggests that about 75% of BiP are being bound to C_{H1} under the chosen experimental conditions (BiP and C_{H1} were used at a ratio of 10 μM : 30 μM)^{10,48}. To test a potential influence of C_{H1} on BiP deAMPylation, BiP^{AMP} was preincubated with C_{H1} and deAMPylated in the presence of AMP-PNP or ADP. The reaction progress was monitored via western blotting (**Fig. 4d**). Similar to the AMPylation reaction, no significant effect of C_{H1} was observed on the deAMPylation of BiP^{AMP} bound to ADP or AMP-PNP, suggesting that neither AMPylation nor deAMPylation of BiP is directly regulated by the presence of unfolded proteins, regardless of the bound nucleotide. Of note, we observed that the ability of AMPylated BiP to bind unfolded protein substrates is impaired compared to unmodified BiP (Supplementary Fig. 12). Together, it seems unlikely that the described minor effects of C_{H1} on BiP AMPylation and deAMPylation are physiologically relevant.

Discussion:

Line 539: While the shift towards the domain-docked conformation upon C_{H1} binding is favorable for FICD mediated BiP-AMPylation, C_{H1}-binding may also lead to rearrangements within the loops or partial obstruction of loop regions within the SBD of BiP that are relevant for the interaction with FICD. We speculate that these two competing processes abrogate each other. The deAMPylation of BiP, however, does not seem to be directly regulated by unfolded proteins since its ability to bind unfolded protein substrates is impaired. Our observation is in line with previous results, demonstrating that the modified species of BiP is not complexed to protein substrates.⁵⁹

59. Hendershot, L. M., Ting, J. & Lee, A. S. Identity of the Immunoglobulin Heavy-Chain-Binding Protein with the 78,000-Dalton Glucose-Regulated Protein and the Role of Posttranslational Modifications in Its Binding Function. Mol. Cell. Biol. 8, 4250–4256 (1988).

Methods:

Line 711:

Analytical HPLC experiments

To analyze the binding of C_{H1} to BiP, C_{H1} was first labeled with NHS-Fluorescein (#46410, Thermo Scientific) according to the manufacturer instructions with minor adjustments. The labeling was performed in phosphate buffered saline at pH 7.4 at a molar ratio of 1:1 (C_{H1}: NHS-Fluorescein) for 1 hour at room temperature. Residual label was removed via desalting of C_{H1} into HKM buffer. BiP was incubated for 16 h with C_{H1} in HKM buffer supplemented with 1 mM ADP. Complex formation was analyzed via size exclusion chromatography (Superdex 10/300 200 pg column) at a flow rate of 0.5 mL/min. The absorption was measured at 280 nm and 496 nm.

5. The authors should do a detailed structural comparison of the reported FICD-BiP complex with published Fic-enzymes in complex with their substrates (such as the IbpA-Cdc42 complex). It seems that the TPR-like motifs of IbpA also interacts with Cdc42. Are there any similarities?

As pointed out by the reviewer, target recognition of IbpA relies on its adjacent domain (referred to as the arm domain, which is structurally distinct from TPR-domains). Inspired by the comment, we superimposed the Fic motifs of both FICD and IbpA to illustrate the overall arrangement of Fic -and TPR/arm -domains and highlighted differences in the arm/TPR based interfaces in the IbpA:Cdc42 and FICD:BiP complex. We observed that 1) the adjacent arm and TPR domain of IbpA and FICD differ in their relative position to the Fic-domain and 2) in contrast to the TPR interface in FICD:BiP, the interaction of the arm domain of IbpA with Cdc42 relies on an extensive hydrophobic interface.

As suggested, we included these observations in a supplementary figure and extended the discussion in this regard.

Supplementary Information:

Supplementary Fig. 14

Supplementary Fig. 14. Comparison of the IbpA:Cdc42 complex with the FICD:BiP complex. The Fic-motifs (red) of IbpA in IbpA:Cdc42 (PDB: 4ITR) (grey:white) and FICD in FICD:BiP (green:wheat) were superimposed to highlight the relative orientation of the adjacent arm domain of IbpA and the TPR domain of FICD. For clarity, only the most prominent interacting residues are highlighted in the detailed comparisons of the TPR-based and arm-domain based interfaces (circles). The hydrophobic interfaces are displayed as yellow pyramids.

Discussion:

Line 477: Interestingly, superimposition of the Fic motif in IbpA:Cdc42 and FICD:BiP reveals that the adjacent arm and TPR domain of IbpA and FICD differ in their relative position to the Fic-domain (Supplementary Fig. 14). In contrast to the TPR interface in FICD:BiP, the interaction of the arm domain of IbpA with Cdc42 relies on an extensive hydrophobic interface.

minor concerns:

1. For the crystallographic data, the I/σ for the highest resolution shell is 1.1 with very high Rmerge (0.982) and low CC1/2 (0.48), which means the data in this shell is almost meaningless. I would suggest cutting the resolution to a more meaningful limit with reasonable I/σ , Rmerge and CC1/2.

The choice of the resolution cut-off was guided by criteria that primarily aimed at optimal map quality and resolution statistics. Generally, a CC1/2 value close to 50% is considered as a reasonable cut-off criterion. We agree that the I/σ for the highest resolution shell is relatively low, yet still in an acceptable range given the anisotropy of our diffraction data. The chosen cut-off allowed us to avoid excluding possibly useful reflections in the best diffracting direction (Along l axis) as indicated by the AIMLESS assessment of the uncut dataset. Eventually, the quality of the map and the resulting solid refinement statistics as shown in the Supplementary Table 1 and wwPDB X-ray Structure Validation Summary Report encouraged us to think we have taken the right decision in this respect.

AIMLESS log data:

Resolution of input data: 1.64Å, resolution estimate 2.42Å (2.62Å from I/σ)

Anisotropic limits: - In h k plane 2.66Å CC(1/2), 2.79Å I/σ - Along l axis 2.10Å CC(1/2), 2.29Å I/σ

2. For the predicted FICD-BiP-SBD contacts by MD, E514 is on beta7, not on loop78.

We thank the reviewer for pointing this out and clarified the sentence.

Line 344: The BiP-substitutions E514A_{BiP} in beta-sheet 7 and K516A_{BiP} in loop 7,8 (L_{7,8}) impaired AMPylation by approximately 88% and 99%, respectively, whereas K521A_{BiP} (in L_{7,8}) and R492A_{BiP} (in L_{5,6}) did not notably affect AMPylation.

Reviewer #3

The manuscript entitled “Specificity of AMPylation of the human chaperone BiP is mediated by TPR motifs of FICD” by Fauser provides valuable structural insight into the recognition of substrates by a variant of the dual AMPylating/DeAMPylating enzyme FICD. Using thiol-reactive nucleotide derivatives, the authors were able to solve the crystal structure of a mutant AMPylation competent FICD in complex with its substrate BiP. The author’s structure reveal interaction interfaces between the FICD TPR domains and identified several key interactions for substrate recognition.

Overall, this is an important step that provides valuable insight into the substrate recognition by Fic, albeit using a restricted variant FICD. However, a few additional experiments would clarify this observation and increase the significance of their findings.

We thank the reviewer for the positive reception and in particular for the detailed analysis of the figures and input on improving them.

Remaining questions and concerns:

1) Several mutations were present in the FICD construct used for crystallization (FICD 102-445, E234G, E404C, T168A, T183A, L258D). The mutations E234G and L258D have known effects on the AMPylation and deAMPylation activity of FICD. Do mutations in the AMPylation sites T168 and T183 have effects in FICD enzymatic activity?

We tested the both FICD T168A T183A L258D and FICD T168A T183A for their AMPylation and deAMPylation activity (see below). We observed that substitution of the automodified threonines by alanines leads to decreased AMPylation and deAMPylation activity. However, whether these effects result from structural perturbation of FICD T168A T183A L258D / FICD T168A T183A or from the absence of a possibly stimulating effect of automodification cannot be concluded here. Since these mutations were only introduced for the sake of homogeneity in crystallography, we do not include these results in the manuscript. Furthermore, the analysis of these observations requires elaborate investigation and is out of the scope of this publication.

Figure legend: *Effect of T168_{FICD} and T183_{FICD} on BiP AMPylation and deAMPylation.* **a)** AMPylation of BiP 19-654 WT by FICD L258D (+) and FICD T168A T183A L258D (TA/TA). Experimental conditions as in the manuscript. **b)** DeAMPylation of BiP 19-654 WT by FICD L258D (+) and FICD T168A T183A L258D (TA/TA). Experimental conditions as in the manuscript. **c)** DeAMPylation of BiP 19-654 WT by FICD (+) and FICD T168A T183A (TA/TA). Experimental conditions as in the manuscript.

2) The authors crystalized a monomeric version of FICD. However, they comment that the overall structure of FICD's catalytic domain is not significantly altered with the interaction of BiP's NBD. A current hypothesis in the field is that FICD becomes monomeric in vivo when it's activity switches from deAMPylator to AMPylator. Does the authors crystal structure indicate any structural corroboration for this hypothesis?

Structural analysis of AMPylation-active monomeric FICD and the comparison to its dimeric deAMPylation-active counterpart showed only very subtle changes (Perera et al., 2019). More specifically, in monomeric FICD the side chain of E234 is differently oriented and thus allows productive nucleotide binding. Since we used FICD including a E234G mutation for generation of a stable ternary complex, no conclusion can be drawn on the positioning of E234. Additionally, we observed that in the complex structure the SBD and Fic-domain are partially undocked and the α -phosphate shifted from its AMPylation competent position. Therefore, the solved structure cannot corroborate this hypothesis.

However, Perera et al. (2019) observed that the TPR domain was flipped by 180° in monomeric FICD L258D when bound to ATP or AMP-PNP (Perera et al., 2019). The authors did not speculate on this finding as it likely presents a crystallization artefact. Our complex structure, however, shows that only a minor twist of the TPR motifs and not a major reorientation contributes to BiP binding and AMPylation. Hence, the observed flip in FICD L258D:ATP is likely to be a crystallographic artefact. We included these observations in our discussion as follows:

Line 508: Interestingly, the TPR motifs of monomeric FICD in complex with ATP and AMP-PNP were shown to be flipped by 180 degrees.³¹ The complex structure of FICD and BiP, however, shows that only a minor twist of the TPR motifs and not a major reorientation contributes to BiP binding and AMPylation. Hence, the observed flip in FICD L258D:ATP is likely to be a crystallographic artefact.

3) Figure 3a,b: To determine if the interactions between FICD's TPR domain and BiP were important for substrate recognition, the authors generated several point mutants in the FICD L258D construct and used these double mutants for AMPylation assays. However, FICD L258D itself still retains deAMPylation activity (CITE). Thus, the authors cannot conclude that these mutants have lost their AMPylation activity by simply measuring BiP-AMP by western. It is likely BiP-AMP is being turned over during the assay via deAMPylation. Without establishing that deAMPylation is also being affected, little can be concluded here.

We appreciate the reviewer's comment. For validation we preferred to use the monomeric version of FICD since it likely represents the physiological AMPylator. We are aware of the residual deAMPylation activity of FICD L258D and therefore monitored the kinetics of the AMPylation reaction of FICD L258D via western blot. The linear increase of AMPylated BiP over time indicates that the deAMPylation of BiP is negligible within the observed timeframe (30 min) (see below Supplementary Fig 5). If the alanine substitutions within FICD L258D do not increase the deAMPylation activity compared to FICD L258D, the observed defect in BiP AMPylation (by FICD L258D + alanine substitution) consequently results from a loss in AMPylation activity upon alanine substitution. As requested by the reviewer, we therefore monitored the deAMPylation activity of all FICD L258D mutants and established that mutations within the TPR motif have a similar effect on deAMPylation activity as on AMPylation activity. The mutations within the Fic domain of FICD only mildly decrease the deAMPylation activity but do not enhance the latter.

We included this data to the supplementary information and added a corresponding statement in the results section.

Please note that in deAMPylation assays we compared the decrease of BiP^{AMP} of the FICD mutants. The relative deAMPylation activity describes the normalized difference of BiP^{AMP} with and without (-) enzyme. Some mutants exhibited very low deAMPylation activity (Δ TPR, K124A). Due to the natural distribution of values in western blotting in some replicates slightly negative values were obtained after subtraction. We set these negative values to '0' since negative values are not meaningful in this case and specifically stated this procedure in the figure legend as well as in the methods section. We kindly ask the reviewer for feedback in this regard whether this procedure is acceptable.

Supplementary Figure 5:

Supplementary Fig. 5. AMPylation and deAMPylation by FICD L258D and FICD. **a)** AMPylation of BiP 19-654 by FICD 102-458 L258D visualized by western-blotting using an α AMP-specific antibody. The linear increase of AMPylated BiP over time (until $t = 30$ min) indicates negligible deAMPylation activity under the chosen experimental conditions. **b)** AMPylation of BiP 19-654 by FICD 102-458 E234G ('E234G') and FICD 187-458 E234G ('FICD Δ TPR') visualized by western-blotting using an α AMP-specific antibody. **c)** DeAMPylation of BiP by FICD 102-458 WT ('FICD WT') and FICD 187-458 WT ('FICD Δ TPR WT') visualized by western-blotting using an α AMP-specific antibody. **d)** DeAMPylation of BiP^{AMP} by FICD L258D and corresponding alanine substitutions visualized by western-blotting using an α AMP-specific antibody. The reaction was performed in presence of 0.5 mM ApNHpp (non-transferable ATP analogue) and stopped after 5 min by addition of Laemmli buffer. The relative deAMPylation activity describes the normalized difference of BiP^{AMP} with and without (-) enzyme. Since some mutants exhibited very low deAMPylation activity (Δ TPR, K124A), in some replicates negative values were obtained after subtraction and set to '0'. **e)** DeAMPylation of BiP^{AMP} by FICD and corresponding alanine substitutions visualized by western-blotting using an α AMP-specific antibody. The same experimental procedures as in 'd' were applied.

Line 249: Since HYPE L258D is also able to deAMPylate BiP, we assured that within the observed time period, deAMPylation was negligible and that the introduced alanine substitutions did not enhance deAMPylation activity (Supplementary Fig. 5).

Materials and Methods:

Line 673: In order to produce AMPylated BiP, 6xHis- BiP was incubated for 16 h at 23 °C with FICD 102-458 E234G at a ratio of 1:20 using the already described AMPylation reaction conditions. 6xHis- BiP was affinity purified using the same conditions as described in the section for protein purification and dialyzed twice against 20 mM HEPES-KOH pH 7.4, 150 mM KCl, 1 mM EDTA at 4 °C. Alternatively, AMPylated BiP was produced by incubation with FICD 45-458 E234G at a molar ratio of 1:100 and dialyzed twice against 20 mM HEPES-KOH pH 7.4, 150 mM KCl, 1 mM EDTA at 4 °C.

Line 685: In both AMPylation and deAMPylation assays the reaction progress was determined semi-quantitatively by densitometric analysis. In deAMPylation assays for the comparison of FICD (L258D) with various alanine substitutions, the relative deAMPylation activity was determined as the normalized difference of BiP^{AMP} with and without (-) enzyme. Some mutants exhibited very low deAMPylation activity which, due to the natural distribution of values in western blotting, in some replicates yielded slightly negative values after subtraction. These negative values were set to '0'.

4) Figure 3a and 3f : The authors do not directly address the same interactions between FICD's TPR domains and BiP's NBD are needed for deAMPylation of BiP. The authors should perform deAMPylation assays with the FicD interaction mutants described here. Evidence that FicD does or does not use the same substrate interactions to deAMPylate vs AMPylate would be valuable information in understanding how this enzyme is regulated, and the identification of mutants that further separate these activities is valuable to the field. The

authors touch on this in the discussion (Lines 455). With all the tools in hand, it seems like a straightforward next step in their study.

As suggested, we produced FICD WT bearing the corresponding alanine substitutions and included data on the contribution of FICD residues on deAMPylation of BiP (see Supplementary Figure 5e, previous comment). Overall, we observed that the deAMPylation reaction is more robust than the AMPylation reaction towards alanine substitutions.

We determined that deAMPylation is indeed dependent on the same residues within the TPR motifs which suggests that the TPR motifs are involved in the deAMPylation reaction in a similar fashion as observed for the AMPylation reaction (as demonstrated by crystallography and biochemical validation). Interestingly, the alanine substitutions within the Fic domain retained throughout high deAMPylation activity whereas AMPylation was impaired upon mutagenesis (H318A, R396A, H401A, E404A). Importantly, R396A_{FICD}, which had the most pronounced divergent effect on AMPylation and deAMPylation, was only inhomogeneously obtained after protein purification. We conclude from these observations that the deAMPylation complex may have qualities different to the AMPylation complex regarding the contribution of the Fic domain to BiP binding.

We included and discussed our findings in the results and discussion section as well as in Supplementary Figure 5 (see previous comment).

Line 286: Furthermore, by testing FICD WT and the corresponding alanine substitutions in deAMPylation assays, we observed that deAMPylation is indeed dependent on the TPR motif and its specific interactions that before proved important for BiP AMPylation (Supplementary Fig. 5).

Line 359: Interestingly, the deAMPylation reaction by FICD WT was only mildly affected by all tested alanine substitutions (Supplementary Figure 5). It has to be considered, however, that R396A_{FICD} which in contrast to the deAMPylation reaction proved very important for the AMPylation reaction resulted in inhomogeneous enzyme preparation as judged by size exclusion chromatography during protein purification.

Line 496: While the contribution of the TPR-NBD interface appears to be important for both BiP AMPylation and deAMPylation, the interface of the Fic domain and SBD seems to have different qualities for either reaction as specific alanine substitutions affected the two reactions to a different extent.

5) The authors show that the presence of excess ERdj3 inhibits the AMPylation of BiP by the overactive FICD E234G (Figure 3c) and that presence of excess unfolded substrate, CH1 does not alter AMPylation of deAMPylation of BiP (Figure 4c). However, both of these systems are highly artificial and tell us little about how FICD would interact with a BiP that is cycling in the ER. A more useful system to study would be to compare AMPylation and deAMPylation of BiP in the presence of stoichiometric ERdj3 with increasing concentrations of CH1.

Indeed, the conditions in the above mentioned experiments do not represent the physiological situation. In the ER BiP activity is governed by several nucleotide-exchange factors and ERdjs, a situation that can hardly be reconstructed in an *in vitro* setup. However, the purpose

of the described experiments required the chosen conditions in order not to produce complex effects. In the first experiment we added the J-domain to the AMPylation reaction with the aim to further corroborate our findings on the TPR – NBD interface. We indeed observed a competition of the J-domain with FICD for the NBD binding site. In the second experiment we investigated whether C_H1 binding, which proved to shift the conformational equilibrium of BiP to a domain-docked conformation (Marcinowski et al., 2011), will affect BiP AMPylation. We hence took the therein published procedures as a reference.

6) Figure 4e: The authors found that the TPR domains of FICD are required for AMPylation of EEF1A2. It would be interesting to know if the same residues in the TPR domain required for recognition of BiP (K121, K124) are also required for EEF1A2.

We agree with the reviewer that this would be interesting. In further experiments we observed that monomeric FICD is unable to AMPylate EEF1A2 which suggests a TPR-dependent yet different binding mode. We do not address the interaction of FICD and EEF1A2 in detail, since the mode of interaction would be highly speculative without any experimental basis. As pointed out in our comment to reviewer 1, an amino acid sequence alignment between BiP and EEF1A2 did not provide a hint on the potentially relevant interactions, either.

7) Line 448: The authors make a point to mention that the monomeric FICD has a nearly identical structure that that of previously published dimeric FICD “indicating that the proposed conformational plasticity of the Fic domain is difficult to grasp via crystallography”. In this discussion, the authors do not take into consideration the other mutations they have added to their crystallized construct, namely E234G that dispels a critical salt bridge in the FICD domain active site, resulting in artificial activation of the AMPylation activity and abolishment of the deAMPylation activity of this enzyme . Since others have indicated that the monomeric form of FICD allows this salt bridge to become more malleable, it is not surprising that little changes are observed in the structure where this interaction is not present.

In the discussion we mention that “*the Fic domain of complexed FICD is virtually identical to the Fic-domain of dimeric and monomeric FICD structures*” (Line 505), regardless of substitution of E234 by a glycine. The most prominent change in FICD upon monomerization is the reorientation of the sidechain of E234 (Perera et al., 2019), on which positioning we cannot draw any conclusions since we employed a FICD construct bearing a E234G substitution to prepare a stable AMPylation complex. In agreement with previous reports (Perera et al., 2019), our results confirm that the conformational changes in solution that lead to the reorientation of the E234 side chain upon monomerization require elaborate in-solution structural investigation.

We adjusted the discussion as follows:

Line 500: Recent structural and biochemical studies suggested that monomeric FICD is AMPylation active in contrast to dimeric FICD (AMPylation inactive, which acts as a deAMPyase). While the positioning of phosphates of the bound ATP as well as the side chain of E234 in these structures differ, the Fic-domain does not undergo notable structural changes^{29,31}. Since the structure of complexed FICD bears a E234G mutation, no conclusion can be drawn on the side chain’s positioning. The Fic domain of complexed FICD is virtually

identical to the Fic-domain of dimeric and monomeric FICD structures, indicating that the proposed conformational plasticity of the Fic domain that governs the positioning of E234 side chain is difficult to grasp via crystallography^{23,31}.

8) Furthermore, the authors should acknowledge that FICD is a membrane associated protein and thus its movement and dimerization may show different qualities in vivo.

We thank the reviewer for this remark and therefore highlighted the membrane association of FICD in the discussion and referred to an illustration of FICD's localization and BiP binding in a supplementary figure. Since 1) the monomerization of FICD and its consequences has been investigated and discussed by others (Casey et al., 2017; Perera et al., 2019) and 2) this manuscript focusses on the contribution of the TPR motifs on BiP binding and their functional implications, we feel an extension of the discussion upon monomerization is out of the scope of the manuscript.

Discussion:

Line 483: Importantly, FICD is anchored to the ER membrane via its transmembrane domain.^{15,52} A stretch of more than 50 partially unstructured amino acids (JPred⁵³) between the transmembrane domain and the BiP-engaging TPR motifs would allow the herein proposed mode of interaction of FICD and BiP (Supplementary Fig. 15).

52. Rahman, M. et al. Visual neurotransmission in Drosophila requires expression of Fic in glial capitate projections. *Nat. Neurosci.* **6**, 871–875 (2012).

53. Drozdetskiy, A., Cole, C., Procter, J. & Barton, G. J. JPred4: A protein secondary structure prediction server. *Nucleic Acids Res.* **43**, W389–W394 (2015).

Supplementary Fig. 15

Supplementary Fig. 15. Schematic representation of the FICD:BiP complex associated to the ER membrane. BiP (orange) is bound to FICD (blue). The N-terminal part of FICD is predicted to bear a cytoplasmic tail, followed by a transmembrane domain that anchors the protein to the ER membrane (Uniprot). In between the transmembrane domain and the TPR motif that is involved in BiP recognition are more than 50 amino acids (aa), partially structured as helix, partially unstructured as predicted by JPred.⁵³

9) Line 484: The authors theorize that FICD localization could change from within the ER to cytoplasmic “upon certain stressors or in different cell lines and tissues.” Has such a change in localization of a membrane protein been described before? A reference to such an example would be beneficial in understanding how a such a topological change is possible.

Indeed, such topological changes have been described before. As requested by the reviewer we extended our discussion by adding the corresponding reference.

Line 547:

The finding that the TPR motifs of FICD are crucial for AMPylation of BiP and EEF1A2, but not for the AMPylation of UMPS may provide a means to distinguish between physiological substrates of FICD and unspecific AMPylation events. While EEF1A2 and UMPS reside in the cytosol, FICD was shown to localize within the ER under normal conditions.^{15,60} However, it is currently uncertain whether FICD localization changes (e.g. by regulated alternative translocation⁶¹) upon certain stressors or in different cell lines and tissues. More detailed knowledge of FICD’s localization would greatly help to understand FICD’s role in a physiological context. Still, it remains possible that other, yet unidentified cytosolic AMPylators, modify described AMPylation substrates such as EEF1A2 and UMPS.

61) Chen, Q. et al. Inverting the Topology of a Transmembrane Protein by Regulating the Translocation of the First Transmembrane Helix. *Mol. Cell* **63**, 567–578 (2016).

10) In addition, recent work has shown that EEF1A2 and UMPS AMPylation were not dependent on the expression of FICD in the cell as loss of FICD did not change observed AMPylation levels of these proteins. This is in contrast to BiP AMPylation which is FICD dependent. (DOI: 10.1038/s41467-019-14235-6) A more likely possibility is another cytosolic enzyme is responsible for AMPylation of EEF1A2 and UMPS which should be addressed in the discussion.

We agree with the reviewer that other yet unidentified enzymes may be responsible for AMPylation of UMPS and EEF1A2. We therefore extended our discussion as shown in the previous question (see above, Reviewer 3, Question 9).

11) Line 440: Multiple AMPylation sites on BiP have been reported by several research groups. The authors comment that when BiP T518A was used as a substrate, no FICDTREND2-BIP ternary complex formation occurred. They argue that this suggests T518 is the only valid BiP AMPylation site. However, BiP T518A has previously been shown to be

not fully functional as it not viable in flies. Importantly the BiP T366A mutation has been found to phenocopy fic knockouts in flies (DOI: 10.7554/eLife.38752).

Furthermore, the authors state that they are also using the BiP 229A mutant which biases BiP to an ATP bound conformation and results in a steric block for access to the T366 site. It is possible that a BiP T518A mutant biases BiP to a conformation that blocks FICD recognition of other AMPylation sites. If AMPylated BiP is used as a substrate, can additional FICDTREND2-BIP ternary complex formations occur? Do FICDTREND2-BIP ternary complex formations occur? If wildtype BiP is used as a substrate, are there multiple populations of FICDTREND2-BIP ternary complex formations?

We observed that ternary complex formation of BiP and FICD results in a homogenous complex that can be purified in multi-milligram quantities (Figure 1f and g). In order to assess whether an additional modification on T366 is present after AMPylation of BiP, we performed an experiment that does not rely on synthesized nucleotide analogues. Therefore, we AMPylated BiP WT by FICD E234G and analyzed if we can detect multiple AMPylation sites in high resolution mass spectrometry. In our hands BiP is only monoAMPylated nearly to completion. Subsequently, we digested monoAMPylated BiP and searched for AMPylated peptides via LC-MS/MS analysis. We only detected peptides bearing a modified T518 residue and only unmodified peptides comprising T366. We added these results as supplementary information and briefly referred to them in the results section.

Supplementary Fig. 2

Supplementary Fig. 2 BiP is monoAMPylated at T518. a) Intact mass spectrometry of unmodified (green) and AMPylated BiP (blue). b) Peptide spectra of trypsin-digested BiP^{AMP} reveal that FICD modifies BiP specifically at T518. Peptides comprising T366 were exclusively found unmodified.

Results:

Line 142: Importantly, ternary complex formation proved dependent on T518_{BiP}, since its mutation to alanine abolished ternary complex formation completely. This observation and mass spectrometric analysis of AMPylated BiP supports T518_{BiP} as the major modification site within BiP as reported before (Supplementary Fig. 2).^{19,25}

Discussion:

Line 488: While AMPylation of BiP has been reported to occur on T166/T366^{14,15} or T518^{19,25}, our results are in favor of T518 as the only relevant modification site, as BiP T518A does not lead to ternary complex formation and only peptides comprising T518 are found to be AMPylated in LC-MS/MS experiments. Furthermore, the covalent linker localizes to T518 in the complex structure and the therein observed contribution of the TPR motifs to BiP binding seems unlikely in the proposed AMPylation of T166/T366.

Methods:

Line 750: High resolution mass spectra of BiP and BiP^{AMP} were recorded on a Bruker maXis IITM QTOF mass-spectrometer coupled to an Bruker Elute LC system (monolithic column Thermo ProSwift RP-4H (50 mm x ID 1 mm), flow: 300 μ L/min, gradient of eluent A: milliQ H₂O + 0.1% formic acid and eluent B: acetonitrile + 0.1% formic acid (5%B to 80%B in 2 min), ionization method: Electrospray Ionization). Samples were injected as 2 μ L from 0.3 mg/mL. Ion spectra were analyzed from the base peak chromatogram (BPC) using Bruker Compass DataAnalysis 5.1 and spectra were deconvoluted using the maximum entropy algorithm.

Acknowledgements:

Line 1039: C.P. synthesized the TReNDs and performed LC-MS of FICD and ternary complex.

12) Line 350: Fig. 4b, BiPAMP in the presence of ADP showed an increase and then decrease of AMPylation. Is this accidental in the figure or a consistent result? This should be discussed.

We replaced the blot with a better representative of the findings to avoid confusion. The blot illustrates that BiP^{AMP}:ADP is deAMPylated only after prolonged incubation with FICD.

Figure 4b

13) Fig 2a: Though noted in the legend, the figure can be labeled for better illustration, including the domains and nucleotides. The linker can be colored differently as currently it is not obvious.

We have improved the illustration according to the reviewer's suggestion.

Figure 2:

14) Line 183: “C α position of M135” can be shown in the Fig. 2c.

The figure and figure caption have been adjusted.

Figure 2c:

Line 234 (Figure legend): The distance of C α of M135_{FICD} (red contoured spheres) from the complex and isolated FICD is calculated to 9.8 Å.

15) Line 196-197: FICD also interacts with SBD as shown in Fig. 2b. This should be mentioned.

This is mentioned a few lines below.

Line 210: In addition, P444_{BiP} from the SBD interacts with K134_{FICD} and M135_{FICD}.

16) Fig. S4b: The time point “60” should be “1” if “t in h” is the unit.

This oversight has been corrected in the revised version of the manuscript.

Figure S5c (previously S4b):

17) Line 276: What is “QPD”? This is not explained and confusing.

We extended the explanation on the abbreviations.

Line 297: Furthermore, we used both ERdj3 WT (containing the for functional BiP interaction essential HPD-motif) and the ATPase-stimulation-deficient ERdj3 H53Q (containing a mutated form of the HPD motif: QPD) as an additional control⁴⁴.

18) Supplemental figures should be cited more accurately including “a, b, c...”. It’s hard to find the exact figure panels the authors refer to as there are many in each figure and they are not cited in sequential numbers.

We understand the reviewers concern but need to adhere to the Nature Communications guidelines which do not allow citing Supplementary Figures with ‘a,b,c...’. However, as suggested, we rearranged some of the figures to be able to cite them sequentially. Hence, parts of former Fig. S3 were used as separate Supplementary Figures (Supplementary Fig. S6 and S16) or were included in other Supplementary Figures (Supplementary Fig. S12).

Supplementary Fig. 6

Supplementary Fig. 6 Crystal contact of Y172_{FICD} with the nucleotide in the nucleotide binding pocket of BiP. Detailed view on crystal contacts of R297_{BiP}, Y172_{FICD} and AMP-PNP, bound to BiP. FICD is colored purple and BiP is colored wheat.

Supplementary Fig. 12

Supplementary Fig. 12. Binding of BiP to the unfolded protein substrate C_H1 is impaired upon AMPylation. **a)** Left panel: Detailed view on the SBD of complexed BiP (orange) overlaid to BiP:ATP (green, PDB: 5E84). The cleft where protein substrates bind is open and indicated by the red star. Right panel: Position of the substrate binding cleft in the complex structure. The cleft where protein substrates bind is indicated by the red star. **b)** Binding of unmodified BiP (10 μ M) to the unfolded protein substrate C_H1 (2.5 μ M). The upper panel displays BiP trace at 280 nm whereas the lower panel indicates BiP binding to FITC labeled- C_H1 at 496 nm. **c)** Binding of AMPylated BiP (BiP^{AMP}) (10 μ M) to the unfolded protein substrate C_H1 (2.5 μ M). The upper panel displays BiP^{AMP} trace at 280 nm whereas the lower panel indicates BiP^{AMP} binding to FITC-labeled C_H1 at 496 nm.

Supplementary Fig. 16

Supplementary Fig. 16. Model of a putative deAMPylation complex. FICD of the complex structure was overlaid on each chain of dimeric FICD (PDB: 4U04).

19) The rotation signs (Fig. 2c and Fig. S3g) are confusing, as they can be interpreted from both directions. Instead a complete oval with an arrowhead should be used with partial of the straight line hid.

This has been adjusted in the illustrations accordingly.

Figure 2c:

Figure S4b:

Figure S16 (previously S3g):

General remarks:

We excluded BiP Q496A from the biochemical experiment (Fig 3f), since the data on characterization (CD, melting temperature, ATPase kinetics) of this particular BiP mutant was incomplete and its meaning for the manuscript limited.

We adjusted the representation of the bar graphs to the journal's specifications with representation of the measured value of each replicate.

Additionally, we exchanged the phostag gel in the lower panel of Fig 1d, illustrating FICD^{TReND} formation, by a better representative.

For clarity reasons we restructured and adjusted the methods section on LC-MS/MS.

Line 765:

LC-MS/MS acquisition, data analysis, and processing

Chromatographic separation of peptides was achieved by nano UPLC (nanoAcquity system, Waters) with a two buffer system (buffer A: 0.1% FA in water, buffer B: 0.1% FA in ACN). Attached to the UPLC was a peptide trap (180 μm x 20 mm, 100 \AA pore size, 5 μm particle size, Symmetry C18, Waters) for online desalting and purification followed by a 25 cm C18

reversed-phase column (75 μm x 200 mm, 130 \AA pore size, 1.7 μm particle size, Peptide BEH C18, Waters). Peptides were separated using a 60 min gradient with increasing ACN concentration from 2% - 30% ACN. The eluting peptides were analyzed on a Quadrupole Orbitrap hybrid mass spectrometer (QExactive, Thermo Fisher Scientific). Here, the 12 most intense ions per precursor scan (1x10⁶ ions, 70,000 Resolution, 120 ms fill time) were analyzed by MS/MS (HCD at 28 normalized collision energy, 1x10⁵ ions, 17,500 Resolution, 50 ms fill time) in a range of 400 – 1300 m/z. A dynamic precursor exclusion of 20 s was used.

LC-MS/MS were searched with the Sequest algorithm integrated in the Proteome Discoverer software (v 2.0, Thermo Fisher Scientific) against the protein sequence of FICD (102-458, E234G) and a contaminant data protein sequence database. Carbamidomethylation was set as fixed modification for cysteine residues and the oxidation of methionine, phosphoadenosine at serine and threonine and tyrosine, pyro-glutamate formation at glutamine residues at the peptide N-terminus as well as acetylation of the protein N-terminus were allowed as variable modifications. Potential peptides with AMPylation were manually inspected to confirm the presence of characteristic AMP signals at 136.0623 m/z (adenine fragment), 250.0940 m/z (adenosine fragment) and/or 348.0709 m/z (adenosine monophosphate fragment) and only then accepted as a valid peptide AMPylation.

References mentioned in point-by-point response

- Awad, W., Estrada, I., Shen, Y., and Hendershot, L.M. (2008). BiP mutants that are unable to interact with endoplasmic reticulum DnaJ proteins provide insights into interdomain interactions in BiP. *Proc. Natl. Acad. Sci. U. S. A.* *105*, 1164–1169.
- Casey, A.K., Moehlman, A.T., Zhang, J., Servage, K.A., Krämer, H., and Orth, K. (2017). Fic-mediated deAMPylation 1 Fic-mediated deAMPylation is not dependent on homodimerization and rescues toxic AMPylation in flies. *J. Biol. Chem.* *292*, 21193–21204.
- Chang, Y.-W., Sun, Y.-J., Wang, C., and Hsiao, C.-D. (2008). Crystal Structures of the 70-kDa Heat Shock Proteins in Domain Disjoining Conformation *.
- Feige, M.J., Groscurth, S., Marcinowski, M., Shimizu, Y., Kessler, H., Hendershot, L.M., and Buchner, J. (2009). An Unfolded CH1 Domain Controls the Assembly and Secretion of IgG Antibodies. *Mol. Cell* *34*, 569–579.
- Hendershot, L.M., Ting, J., and Lee, A.S. (1988). Identity of the Immunoglobulin Heavy-Chain-Binding Protein with the 78,000-Dalton Glucose-Regulated Protein and the Role of Posttranslational Modifications in Its Binding Function. *Mol. Cell. Biol.* *8*, 4250–4256.
- Kityk, R., Kopp, J., and Mayer, M.P. (2018). Molecular Mechanism of J-Domain-Triggered ATP Hydrolysis by Hsp70 Chaperones. *Mol. Cell* *69*, 227–237.
- Marcinowski, M., Höller, M., Feige, M.J., Baerend, D., Lamb, D.C., and Buchner, J. (2011). Substrate discrimination of the chaperone BiP by autonomous and cochaperone-regulated conformational transitions. *Nat. Struct. Mol. Biol.* *18*, 150–159.
- Mayer, M., Reinstein, J., and Buchner, J. (2003). Modulation of the ATPase cycle of BiP by peptides and proteins. *J. Mol. Biol.* *330*, 137–144.
- Mayer, M.P., Laufen, T., Paal, K., McCarty, J.S., and Bukau, B. (1999). Investigation of the interaction between DnaK and DnaJ by surface plasmon resonance spectroscopy. *J. Mol. Biol.* *289*, 1131–1144.
- Perera, L.A., Rato, C., Yan, Y., Neidhardt, L., McLaughlin, S.H., Read, R.J., Preissler, S., and Ron, D. (2019). An oligomeric state- dependent switch in the ER enzyme FICD regulates AMPylation and deAMPylation of BiP. *EMBO J.* *38*, 1–24.
- Preissler, S., Rohland, L., Yan, Y., Chen, R., Read, R.J., and Ron, D. (2017). AMPylation targets the rate-limiting step of BiP's ATPase cycle for its functional inactivation. *Elife* *6*, 1–28.
- Suh, W.C., Lu, C.Z., and Gross, C.A. (1999). Structural features required for the interaction of the Hsp70 molecular chaperone DnaK with its cochaperone DnaJ. *J. Biol. Chem.* *274*, 30534–30539.

REVIEWER COMMENTS

Reviewer #1 (Remarks to the Author):

The authors have thoroughly addressed the comments and improved the manuscript. The revised version is suitable for publication. It is an interesting work.

Reviewer #2 (Remarks to the Author):

The revised manuscript has addressed some of the concerns from the previous review especially the nice AMPylation results on Hsp70 and Hsc70. However, concerns on using ERdj3 HPD (concern 3#) and substrate CH1 (concern 4#) remain. Moreover, the extremely high Rmerge (0.982) and low CC1/2 (0.48) for the crystallographic data is a concern and compromised the data quality.

It is well-established that the HPD of Hsp40s is not sufficient although is required for interacting with Hsp70s. The authors referenced the DnaK-DnaJ structure by the Mayer's group. However, that structure used a fused DnaK-DnaJ construct and the interaction was forced by the fusion and the extremely high protein concentration of the crystallization condition. For the biochemical assay in this manuscript, the author should use a functional ERdj3 construct to avoid misleading results.

In term of the CH1 substrate experiment, it is good that the authors showed that the BiPAMP does not bind the CH1 substrate significantly using sizing column. However, this result seems to make the deAMPylation test meaningless. Moreover, the authors only tested ADP and AMP-PNP. The authors should also test ATP.

Reviewer #3 (Remarks to the Author):

The authors have overall satisfied many concerns with the exception of two. It is our opinion that the overstatement of these results and conclusions could propagate unsupported theories as fact that have not rigorously tested. This leads readers to believe there is merit to these hypotheticals and will allow further reference and embellish weak studies as facts. The text needs to provide an objective analysis of what is actually known based on rigorous and reliable findings.

Point 1

First, we object with the authors decision to omit the activity data of the proteins used in crystal studies. Their mutations change the activity of Fic and this is relevant for accurate interpretation and is scientifically relevant for readers. Knowledge of this change in activity should not be omitted from the publication as it may misinform readers. Without this data, the authors lack transparency on the true nature of their structures. This is especially important since these mutations alter the activity in an unknown way and “cannot be concluded here”.

Point 9 & 10

Second, the authors cite an example of ceramide-induced topology inversion of a transmembrane protein as an example. It one example for a binding protein, not for an enzyme, and requires treatment with ceramide. Based on this and unsubstantiated observations, they are proposing that an enzyme working in the oxidized environment of the ER will act the same way in the reduced environment of the cytosol. Do the authors propose this as specific mechanism? What is the evidence this is relevant discussion for Fic? Additional examples of other induced topological changes should be added to validate this mechanism.

Finally, their answer to this aforementioned statement does not explicitly address point 10.

“The following needs to be included as there is data suggesting that Fic does work on BiP and not other AMPylated proteins.”

The authors should state:

“In addition, recent work has shown that EEF1A2 and UMPS AMPylation were not dependent on the expression of FICD in the cell as loss of FICD did not change observed AMPylation levels of these proteins. This is in contrast to BiP AMPylation which was decreased with the loss of FICD and is therefore FICD dependent. (DOI: 10.1038/s41467-019-14235-6) A more likely possibility is another cytosolic enzyme is responsible for AMPylation of EEF1A2 and UMPS and supporting this proposal is that more enzymes are being found to carry out this chemistry, such as SelO.”

Reviewer #1

The authors have thoroughly addressed the comments and improved the manuscript. The revised version is suitable for publication. It is an interesting work.

We thank the reviewer for the positive reception.

Reviewer #2

The revised manuscript has addressed some of the concerns from the previous review especially the nice AMPylation results on Hsp70 and Hsc70. However, concerns on using ERdj3 HPD (concern 3#) and substrate CH1 (concern 4#) remain.

Moreover, the extremely high Rmerge (0.982) and low CC1/2 (0.48) for the crystallographic data is a concern and compromised the data quality.

We respect the reviewer's comment. Since the implementation of the reviewer's suggestions did not lead to any significant change in electron density and the model, especially in the area of interest for this study, we accept the reviewer's suggestion and set the resolution limit to a lower cut-off (2.64 Å). The statistics have been updated in Supplementary Table 1 accordingly (see below). We adapted the following statements in the results section and recalculated the OMIT map.

Line: 106: In order to gain insights into the fundamentals of FICD mediated BiP-AMPylation that directly regulates ER homeostasis, we solved the atomic structure of the FICD:BiP complex by X-ray crystallography at 2.6 Å.

Line 186: The Fic domain of FICD in the complex structure does not undergo notable structural changes upon interaction with BiP when compared to previously published structures of isolated FICD as illustrated by an overall root mean square deviation (RMSD) of 0.807 Å (aligned residues: 104-434) (Fig. 2c)³¹.

Line 191: Accordingly, the alignment of the isolated and complexed Fic domains yielded an even lower RMSD of 0.648 Å (aligned residues: 187-434).

Line 199: Indeed, with an overall RMSD of 0.636 Å, the complexed BiP is virtually identical to the structure of AMPylated BiP, which also features a domain-docked conformation (Fig. 2d)²⁰.

Line 228: **Fig. 2. Structure of the covalent complex of FICD and BiP.** a) Crystal structure of the covalently linked complex of FICD and BiP solved at 2.6 Å.

Supplementary Fig.4d

Supplementary Table 1

Data collection and refinement statistics.

Data collection

Wavelength (Å)	0.976250
Resolution range (Å)	83.47 - 2.64 (2.77 - 2.64)
Space group	P 4 ₁
Unit cell dimensions (Å)	83.47 83.47 169.48 90 90 90
Total reflections	127291 (17378)
Unique reflections	34024 (4518)
Multiplicity	3.7 (3.8)
Completeness (%)	100 (100)
Mean I/σ(I)	10.1 (1.9)
Wilson B-factor	59.6
R _{merge}	0.057 (0.567)
CC _{1/2}	0.99 (0.72)

Refinement and model statistics

R _{work}	0.192
R _{free}	0.235
Number of non-hydrogen atoms:	6858
macromolecules	6718
ligands	66
solvent	74
RMSD bonds (Å)	0.002
RMSD angles (°)	0.400
Ramachandran:	
Ramachandran favored (%)	98.5
Ramachandran allowed (%)	1.5
Ramachandran outliers (%)	0.0
Average B-factor (Å ²)	73.5
macromolecules	73.8
ligands	65.0
solvent	53.3

It is well-established that the HPD of Hsp40s is not sufficient although is required for interacting with Hsp70s. The authors referenced the DnaK-DnaJ structure by the Mayer's group. However, that structure used a fused DnaK-DnaJ construct and the interaction was forced by the fusion and the extremely high protein concentration of the crystallization condition. For the biochemical assay in this manuscript, the author should use a functional ERdj3 construct to avoid misleading results.

We agree with the reviewer that our results could be misleading since the used constructs, the J domain of ERdj3 and BiP T229A, are only very weakly interacting.

The experiment was designed to further support the data on the $\text{TPR}_{\text{FICD}}:\text{NBD}_{\text{BiP}}$ interface that was obtained by crystallography as well as by our mutagenesis and cell biological studies. Initially, we produced full-length ERdj3 for this purpose. For unknown reasons, we surprisingly observed that full-length ERdj3 did not stimulate ATP hydrolysis under the tested conditions. Therefore, we decided to perform this experiment with the truncated version of ERdj3, which stimulated ATP hydrolysis as expected. The well-characterized binding mode of the J-domain to Hsp70s encouraged us to proceed with the experiment with this truncated version of ERdj3. Moreover, since FICD is AMPylating predominantly BiP:ATP, we had to use the hydrolysis deficient version of BiP (T229A) to avoid ERdj3 stimulated conversion of BiP:ATP to BiP:ADP. Otherwise, it would be difficult to discern whether the AMPylation reaction is influenced by competition for BiP binding or the nucleotide state of BiP.

Considering the reviewer's comment and our rationale for this experiment, we understand the reviewer's concern that the chosen experimental conditions can lead to confusion. Since the hydrolysis deficient BiP T229A would be required for this experiment, we came to the conclusion that the results of this experiment might be misleading regardless of the employed full-length or truncated ERdj3 construct. Therefore, for the sake of clarity, we now decided to exclude this experiment from the manuscript, since it represents merely a supporting experiment to our mutagenesis and cell biological studies. Specifically, the panel in Fig 3c and the corresponding explanations in the results and methods section is removed.

In term of the CH1 substrate experiment, it is good that the authors showed that the BiPAMP does not bind the CH1 substrate significantly using sizing column. However, this result seems to make the deAMPylation test meaningless. Moreover, the authors only tested ADP and AMP-PNP. The authors should also test ATP.

We set out to comprehensively investigate the effect of unfolded proteins on both AMPylation and deAMPylation and demonstrated via AMPylation and deAMPylation assays that unfolded proteins do not largely affect neither AMPylation nor deAMPylation. Encouraged by the reviewer's comment in the first revision, we performed additional control experiments to verify the binding of $\text{C}_{\text{H}}1$ to BiP and BiP^{AMP} . Via analytical size-exclusion chromatography (aSEC) we observed that the ability of BiP^{AMP} to bind $\text{C}_{\text{H}}1$ is impaired compared to unmodified BiP.

As pointed out by the reviewer, the deAMPylation test together with the binding assay now appears redundant. However, since aSEC represents a non-equilibrium method to assess binding, performing the deAMPylation assay to directly observe the consistent effect of unfolded proteins was valuable. Therefore, the deAMPylation assay represents a complementary and functional assay that further supports the results obtained via aSEC. Hence, we decided to transparently show all the obtained

data on this matter as part of a comprehensive analysis of the effect of unfolded proteins on FICD mediated AMPylation and deAMPylation of BiP.

We acknowledge the reviewer's suggestion to also test ATP in regard to deAMPylation of BiP^{AMP}. We tested deAMPylation of BiP^{AMP}:ATP before, yet the analysis of the experiments was not possible for the following reason: In order to produce AMPylated BiP, the hyperactive version of FICD (E234G) is required. After AMPylation, BiP^{AMP} was isolated, however, trace amounts of FICD E234G persistently are carried over. Thus, the ATP in the deAMPylation assay (as suggested by the reviewer) causes immediate reAMPylation of BiP, thus interfering with the deAMPylation assay. This is the reason, why we used AMP-PNP instead of ATP, because FICD E234G cannot accept the non-hydrolyzable ATP analogue as a cosubstrate for (re)AMPylation. AMP-PNP is widely accepted as a non-hydrolyzable ATP surrogate when the use of ATP interferes with the outcome of the experiment. We hence believe that the results of our experiments support the conclusions.

Reviewer #3

The authors have overall satisfied many concerns with the exception of two. It is our opinion that the overstatement of these results and conclusions could propagate unsupported theories as fact that have not rigorously tested. This leads readers to believe there is merit to these hypotheticals and will allow further reference and embellish weak studies as facts. The text needs to provide an objective analysis of what is actually known based on rigorous and reliable findings.

Point 1

First, we object with the authors decision to omit the activity data of the proteins used in crystal studies. Their mutations change the activity of Fic and this is relevant for accurate interpretation and is scientifically relevant for readers. Knowledge of this change in activity should not be omitted from the publication as it may misinform readers. Without this data, the authors lack transparency on the true nature of their structures. This is especially important since these mutations alter the activity in an unknown way and "cannot be concluded here".

As suggested by the reviewer in the first revision, we performed the experiments and attached them to our previous point-by-point response. Here, we acknowledge the reviewer's concern and thus happily present the data in the supplementary material of the manuscript.

Line 153: Of note, alanine substitutions of the autoAMPylation sites slightly reduced both AMPylation and deAMPylation activity of FICD (Supplementary Fig. 3).

Supplementary Fig. 3

b

Supplementary Fig. 3. FICD 102-458 E234G is autoAMPylated at T168 and T183. a) ... b) Effect of T168FICD and T183FICD on BiP AMPylation and deAMPylation. AMPylation of BiP 19-654 WT by FICD L258D (+) and FICD T168A T183A L258D (TA/TA) (left panel). DeAMPylation of BiP 19-654 WT by FICD L258D (+) and FICD T168A T183A L258D (TA/TA) (middle panel). DeAMPylation of BiP 19-654 WT by FICD (+) and FICD T168A T183A (TA/TA) (right panel). Please note, that the substitution of the automodified threonines by alanines leads to decreased AMPylation and deAMPylation activity. The reason for these effects is unclear and may result from structural perturbation of FICD T168A T183A L258D / FICD T168A T183A or the absence of a possibly stimulating effect of automodification.

Point 9 & 10

Second, the authors cite an example of ceramide-induced topology inversion of a transmembrane protein as an example. It one example for a binding protein, not for an enzyme, and requires treatment with ceramide. Based on this and unsubstantiated observations, they are proposing that an enzyme working in the oxidized environment of the ER will act the same way in the reduced environment of the cytosol. Do the authors propose this as specific mechanism? What is the evidence this is relevant discussion for Fic? Additional examples of other induced topological changes should be added to validate this mechanism.

In the discussion, we mention that it is currently uncertain whether FICD changes its localization which makes it difficult to reason whether the reported cytosolic targets may be physiologically relevant or not. As suggested previously by the reviewer, we included a reference to demonstrate that in principle topological changes of membrane proteins are possible. We did not, however, intend to propose the particular mechanism for FICD that is presented in the referenced publication. In order to avoid confusion, we now removed the corresponding reference from the discussion (see below).

Since at the current time, there is no evidence that FICD changes its localization upon certain stressors or in different cell lines and tissues, we extended the discussion as suggested by the reviewer in the next comment.

Finally, their answer to this aforementioned statement does not explicitly address point 10.

“The following needs to be included as there is data suggesting that Fic does work on BiP and not other AMPylated proteins.”

The authors should state:

“In addition, recent work has shown that EEF1A2 and UMPS AMPylation were not dependent on the expression of FICD in the cell as loss of FICD did not change observed AMPylation levels of these proteins. This is in contrast to BiP AMPylation which was decreased with the loss of FICD and is therefore FICD dependent. (DOI: 10.1038/s41467-019-14235-6) A more likely possibility is another cytosolic enzyme is responsible for AMPylation of EEF1A2 and UMPS and supporting this proposal is that more enzymes are being found to carry out this chemistry, such as SelO.”

We acknowledge the reviewer’s suggestion and, for clarity, extend the discussion as suggested with modifications, after correspondence with the author of the mentioned paper (Pavel Kielkowski et al., DOI: 10.1038/s41467-019-14235-6).

Line 551: While EEF1A2 and UMPS reside in the cytosol, FICD was shown to localize within the ER under normal conditions^{15,61}. However, it is currently uncertain whether FICD localization changes (e.g. by regulated alternative translocation⁶²) upon certain stressors or in different cell lines and

tissues. While several studies identified EEFA1A2 and UMPS as AMPylation substrates of FICD,²⁴⁻²⁶ recent work has shown that only BiP AMPylation was dependent on the expression of FICD, in contrast to EEFA1A2 and UMPS that were not enriched with an N6-propargyl ATP probe.⁵⁷ It cannot be excluded, that yet unidentified cytosolic enzymes or enzyme classes might be responsible for AMPylation of EEFA1A2 and UMPS. Speculatively, some pseudokinases may reside in the cytosol and possess AMPylation activity. The AMPylation activity of pseudokinases has recently been demonstrated for the mitochondrial pseudokinase SelO.⁵⁸ Thus, even though EEFA1A2 and UMPS can be AMPylated by FICD in vitro, the physiological relevance of this observation remains enigmatic.

⁵⁷ Kielkowski, P. et al. FICD activity and AMPylation remodelling modulate human neurogenesis. *Nat. Commun.* 11, 517 (2020).

⁵⁸ Sreelatha, A. et al. Protein AMPylation by an Evolutionarily Conserved Pseudokinase. *Cell* 175, 809-821.e19 (2018)

Additional comment from the authors:

During the revision process of this manuscript, another study that employs thiol-reactive nucleotide analogues was published. Therefore, we included the reference in the manuscript accordingly.

Line 101: While this concept was initially designed to identify novel targets of Fic-enzymes, it has proven beneficial for structural analyses by stabilizing the interaction of Fic-enzymes and other AMP transferases with their targets.^{26,37,38}

³⁸ Du, J. et al. Rab1-AMPylation by Legionella DrrA is allosterically activated by Rab1. *Nat. Commun.* 12, 1–16 (2021).